# Function-Based Rhizosphere Assembly along a Gradient of Desiccation in the Former Aral Sea

Wisnu Adi Wicaksono,[a] Dilfuza Egamberdieva,[b] Christian Berg,[c] Maximilian Mora,[a] Peter Kusstatscher,[a] Tomislav Cernava,[a] Gabriele Berg[a,d,e]

[a]Institute of Environmental Biotechnology, Graz University of Technology, Graz, Austria
[b]National University of Uzbekistan, Faculty of Biology, Tashkent, Uzbekistan
[c]University of Graz, Institute of Biology, Graz, Austria
[d]Leibniz Institute for Agricultural Engineering and Bioeconomy, Potsdam, Germany
[e]Institute for Biochemistry and Biology, University of Potsdam, Potsdam, Germany

**ABSTRACT**   The desiccation of the Aral Sea represents one of the largest human-made environmental regional disasters. The salt- and toxin-enriched dried-out basin provides a natural laboratory for studying ecosystem functioning and rhizosphere assembly under extreme anthropogenic conditions. Here, we investigated the prokaryotic rhizosphere communities of the native pioneer plant *Suaeda acuminata* (C.A.Mey.) Moq. in comparison to bulk soil across a gradient of desiccation (5, 10, and 40 years) by metagenome and amplicon sequencing combined with quantitative PCR (qPCR) analyses. The rhizosphere effect was evident due to significantly higher bacterial abundances but less diversity in the rhizosphere compared to bulk soil. Interestingly, in the highest salinity (5 years of desiccation), rhizosphere functions were mainly provided by archaeal communities. Along the desiccation gradient, we observed a significant change in the rhizosphere microbiota, which was reflected by (i) a decreasing archaeon-bacterium ratio, (ii) replacement of halophilic archaea by specific plant-associated bacteria, i.e., *Alphaproteobacteria* and *Actinobacteria*, and (iii) an adaptation of specific, potentially plant-beneficial biosynthetic pathways. In general, both bacteria and archaea were found to be involved in carbon cycling and fixation, as well as methane and nitrogen metabolism. Analysis of metagenome-assembled genomes (MAGs) showed specific signatures for production of osmoprotectants, assimilatory nitrate reduction, and transport system induction. Our results provide evidence that rhizosphere assembly by cofiltering specific taxa with distinct traits is a mechanism which allows plants to thrive under extreme conditions. Overall, our findings highlight a function-based rhizosphere assembly, the importance of plant-microbe interactions in salinated soils, and their exploitation potential for ecosystem restoration approaches.

**IMPORTANCE**   The desertification of the Aral Sea basin in Uzbekistan and Kazakhstan represents one of the most serious anthropogenic environmental disasters of the last century. Since the 1960s, the world's fourth-largest inland body of water has been constantly shrinking, which has resulted in an extreme increase of salinity accompanied by accumulation of many hazardous and carcinogenic substances, as well as heavy metals, in the dried-out basin. Here, we investigated bacterial and archaeal communities in the rhizosphere of pioneer plants by combining classic molecular methods with amplicon sequencing as well as metagenomics for functional insights. By implementing a desiccation gradient, we observed (i) remarkable differences in the archaeon-bacterium ratio of plant rhizosphere samples, (ii) replacement of archaeal indicator taxa during succession, and (iii) the presence of specific, potentially plant-beneficial biosynthetic pathways in archaea present during the early stages. In addition, our results provide hitherto-undescribed insights into the functional redundancy between plant-associated archaea and bacteria.

Address correspondence to Wisnu Adi Wicaksono, wisnu.wicaksono@tugraz.at, Tomislav Cernava, tomislav.cernava@tugraz.at, or Gabriele Berg, gabriele.berg@tugraz.atmailto.

The authors declare no conflict of interest.

**KEYWORDS** Aral Sea, microbiome, desiccation, nutrient cycling, soil microorganisms, revegetation, archaea, bacteria, metagenome

The desertification of the Aral Sea basin in Uzbekistan and Kazakhstan represents one of the most serious anthropogenic environmental disasters of the last century (1). Since the 1960s, the world's fourth-largest inland body of water has been, due to unsustainable expansion of intense, irrigated plant production, especially cotton, under arid conditions, continuously shrinking, which has resulted in an extreme increase of salinity up to 100 g L$^{-1}$ (1–3). In parallel, many hazardous and carcinogenic substances, as well as heavy metals, i.e., Pb, Ni, Cu, and Cd, mainly originating from agricultural effluents and pesticide residues, were deposited in the Aral Sea basin (4–6). Now, these sediments with enriched toxins form carcinogenic salt dust storms that negatively impact environmental as well as human health (1). The high occurrence of toxins not only makes the Aral Sea basin an extreme environment but also further aggravates the survival of humans, especially children (7). The environmental tragedy was documented in the UNESCO historical documents and often studied, especially considering toxicology and human health issues; however, local microbiomes remained mostly unaddressed. The Aral Sea basin nowadays represents a human-made terrestrial desert and unique extremophilic environment. For the past 40 years, primary succession has taken place in the Aral Sea basin by halophytic vegetation (8). Halophytes are known for their potential to improve ecosystem integrity and reduce dust storms (9, 10) and can be used for ecosystem restoration, i.e., by phytoextraction from heavy metal-contaminated soils (11, 12). However, deeper insights into adaption mechanisms of the halophyte holobiont in such environments are essential for targeted restoration approaches.

Plant-associated microbial communities were shown to be essential for plant growth, resistance, and fitness (13–15). This close interaction and dependency of plants on their associated microbes led to the development of the plant holobiont concept (16). Plant microbiome assembly was shown to be driven by various biotic and abiotic factors, i.e., plant genotype, climate, and soil type (15, 17–19). A growing body of evidence showed that host plants can modulate their root microbiota assembly to mitigate stress responses (20–22). Moreover, from intervention studies, we know that microbial inoculants have a much better effect under stress conditions (23, 24). However, despite the important roles of the rhizosphere microbiome in plant nutrient acquisition, stress tolerance, and pathogen suppression, we still lack a holistic understanding of how host plants recruit beneficial microbes under external stress and which taxa they select (25). In general, for rhizosphere assembly, horizontal and vertical microbiota transmission has to be considered (26, 27). Most of the recently proposed models suggest selection and enrichment of copiotrophic microorganisms, such as *Bacteroidetes* and *Proteobacteria*, from bulk soil (reviewed in references 25 and 28). Ren and colleagues (29) suggested "functional compensation," which means that the rhizosphere microbiota is assembled to compensate for the functional requirements of the host plant that cannot be completed by itself. Bergna et al. (30) found vertical transmission for plant beneficial bacteria, while horizontal recruiting mediated adaptation to local environmental conditions. A recent meta-analysis including 557 pairs of published 16S rRNA gene amplicon sequences from the bulk soils and rhizosphere in different ecosystems around the world supports this idea (31). Based on functional predictions, enrichment of specific genes, i.e., those involved in organic compound conversion, nitrogen fixation, and denitrification, was revealed in this meta-analysis (31). Although functional rhizosphere assembly was suggested (25, 31), there is no experimental evidence or mechanistic understanding of the process.

The salt- and toxin-enriched dried-out Aral Sea basin provides a natural laboratory for studying microbial community structures and their ecological roles. First insights into the microbial composition revealed the presence of diverse halophilic and novel prokaryotic taxa (32), and their main drivers, i.e., desiccation, salinity, and local plant species (3), but information on microbiome function is missing. Therefore, we investigated

structural and functional characteristics of plant-associated prokaryotic communities in the Aral Sea basin. We selected the common halophyte *Suaeda acuminata* (C.A.Mey.) Moq., which is the first indigenous pioneer plant naturally colonizing this extreme environment (33). Although *S. acuminata* is not a classical pioneer plant, it fulfils this role under the specific extreme conditions in the Aral Sea basin (34). Samples were obtained along a desiccation gradient from areas that dried out 5 years, 10 years, and 40 years ago near the Large Aral Sea's west shoreline. We hypothesized that this halophyte is equipped with an adapted, functional microbiota for survival, which changed along the desiccation gradient. This study focused on bacterial and archaeal communities, which are expected to be the key players below ground under salinated conditions (23, 24, 32). Here, we provide new evidence that in addition to a plant's intrinsic ability to cope with drought and salt stress, microbiome assembly by cofiltering specific taxa with distinct traits could be one mechanism by which plants adapt to extreme conditions.

## RESULTS

**Bacterial and archaeal abundance assessments in the rhizosphere and in soil indicate specific enrichments.** Bacterial 16S rRNA gene abundances ranged between $1.9 \times 10^8$ and $1.2 \times 10^{11}$ copies $g^{-1}$, whereas archaeal 16S rRNA gene abundances ranged between $1.4 \times 10^8$ and $3.5 \times 10^9$ copies $g^{-1}$ (Fig. 1C and D). Bacterial and archaeal abundance remained stable in the rhizosphere along the gradient of desiccation ($P = 0.727$ and $P = 0.429$, respectively). Interestingly, in the rhizosphere, the highest ratio between archaea and bacteria occurred in the areas that were dried out 5 years ago and then gradually decreased along the gradient of desiccation (Fig. 1B). The rhizosphere samples from areas that were dried out 10 and 40 years ago had a higher bacterial abundance than the respective soil samples from the same area (Fig. 1B) ($P = 0.031$ and $P = 0.005$, respectively). In contrast, a higher archaeal abundance in the rhizosphere than in the respective soil compartment was observed only in the areas that were dried out 40 years ago ($P = 0.043$) (Fig. 1D). Overall, bacterial and archaeal abundance remained stable in the rhizosphere, whereas their abundances decreased in the soil from the area that were dried out 40 years ago.

**Desiccation over the years shaped bacterial and archaeal community structures, suggesting specific roles during the revegetation event.** The microhabitat (soil or rhizosphere) affects prokaryotic diversity and community structures. Following the amplicon sequencing approach, the respective rarefaction curves indicated that the sampling size was sufficient to capture the overall bacterial and archaeal diversity (see Fig. S3A and B in the supplemental material). Alpha diversity analysis indicated that the bulk soils had a significantly higher bacterial and archaeal diversity (H′ = 6.0 and H′ = 4.2, respectively) than the rhizosphere (H′ = 5.2, H′ = 3.3; $P < 0.001$). When the soil and rhizosphere data sets were analyzed separately, bacterial and archaeal diversities were similar along the gradient of desiccation (5 versus 10 versus 40 years, $P > 0.05$). The Shannon diversity index based on shotgun metagenomic sequencing approach also indicated a congruent result ($P = 0.429$ and $P = 0.732$, respectively; Fig. S4). The microhabitat also affected the bacterial and archaeal community structures ($R^2 = 16.4\%$, $P = 0.001$, and $R^2 = 10.8\%$, $P = 0.001$, respectively).

Changes in bacterial and archaeal diversity were observed between bulk soil and rhizosphere along the gradient of desiccation. When soil and rhizosphere samples from the same area were compared, bacterial diversity was lower in the rhizosphere samples than bulk soil samples from areas that were dried out for 5 and 10 years ($P < 0.05$) (see Table S2 in the supplemental material). Interestingly, archaeal diversity was lower in the rhizosphere samples than bulk soil samples from areas that were dried out for 10 and 40 years ($P < 0.05$) (Table S2). These findings might have been due to a trend of decreasing archaeal diversity in the rhizosphere along the gradient of desiccation (Fig. 1D). Gradient of desiccation was also the major factor shaping bacterial and archaeal community structures ($R^2 = 24.6\%$, $P = 0.001$, and $R^2 = 24.2\%$, $P = 0.001$, respectively) (Table S3; Fig. S5A and S5B). By using bacterial and archaeal community profiles that were estimated using Kraken2, a similar pattern was observed

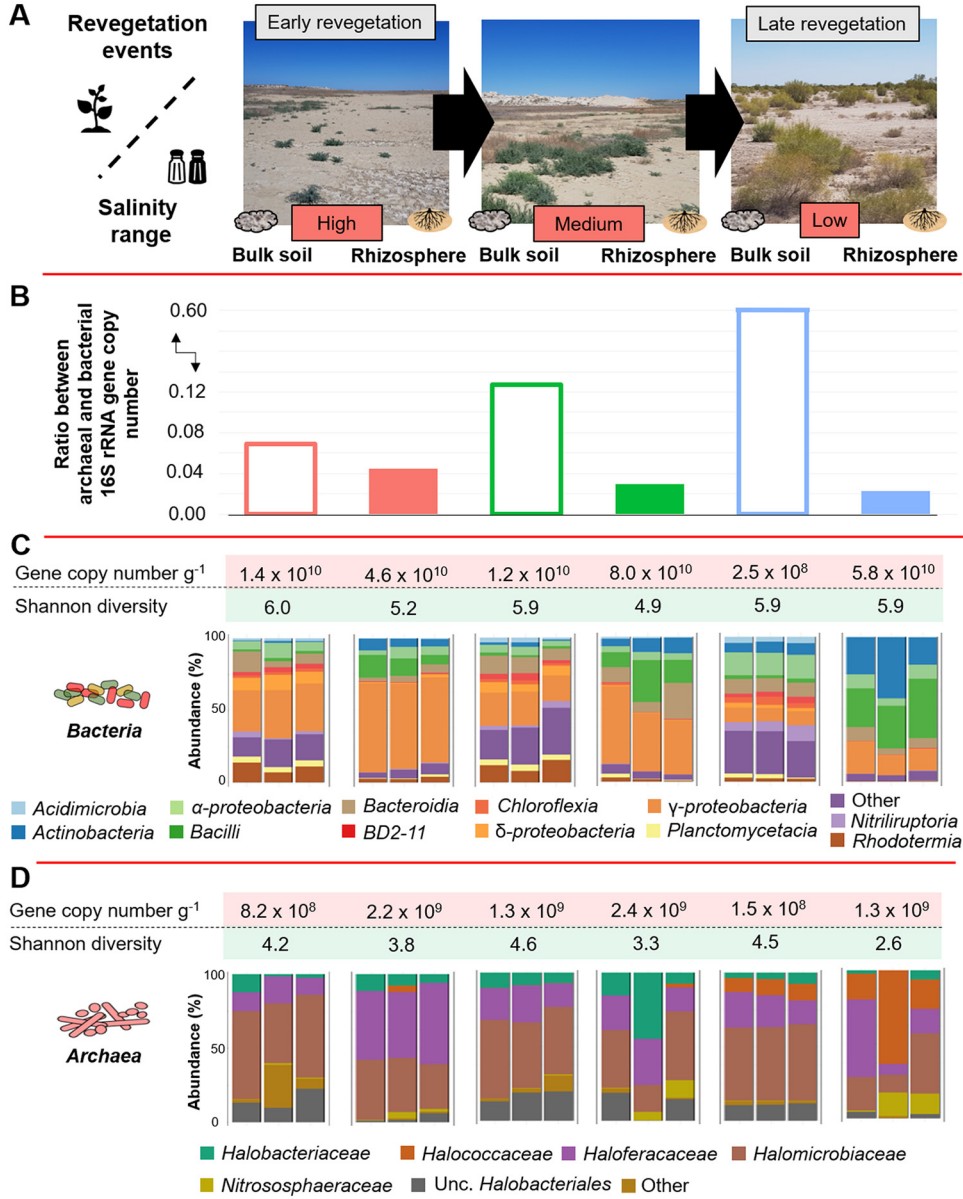

**FIG 1** Sampling site, bacterial and archaeal abundance, diversity, and community structure in bulk soil and the rhizosphere of *Suaeda acuminata*. Different sampling sites represent a gradient of salinity (high, medium, and low) and natural revegetation events in the Aral Sea basin where bulk soil and rhizosphere samples were collected (A). Geochemistry, mineralogy, and the number of visible plants species were previously described (3). Bacterial and archaeal 16S rRNA gene copy numbers were calculated by using qPCR (B, C, and D). The diversity of bacterial (C) and archaeal (D) communities was estimated using the Shannon index in bulk and rhizosphere soils within the analyzed desiccation gradient (5 to 40 years). The assessment is based on amplicon sequencing data.

with the shotgun metagenomic data set (for bacteria, $R^2 = 65.7\%$ and $P = 0.001$; for archaea, $R^2 = 49.9\%$, $P = 0.001$). Together with the quantitative-PCR (qPCR) data, our findings suggest that plants recruit a subset of bacterial and archaeal taxa from the surrounding soil along the gradient of desiccation and maintain certain taxa in high abundance in the rhizosphere.

Comparable to the qPCR results, the ratio between the archaeal and the bacterial relative abundance, determined using amplicon sequencing, gradually decreased in the rhizosphere along the gradient of desiccation (Fig. S2A). A similar pattern was observed with the metagenome data set (Fig. S2B). This finding indicates that bacteria have outcompeted archaea in the late revegetation event. Microbial community

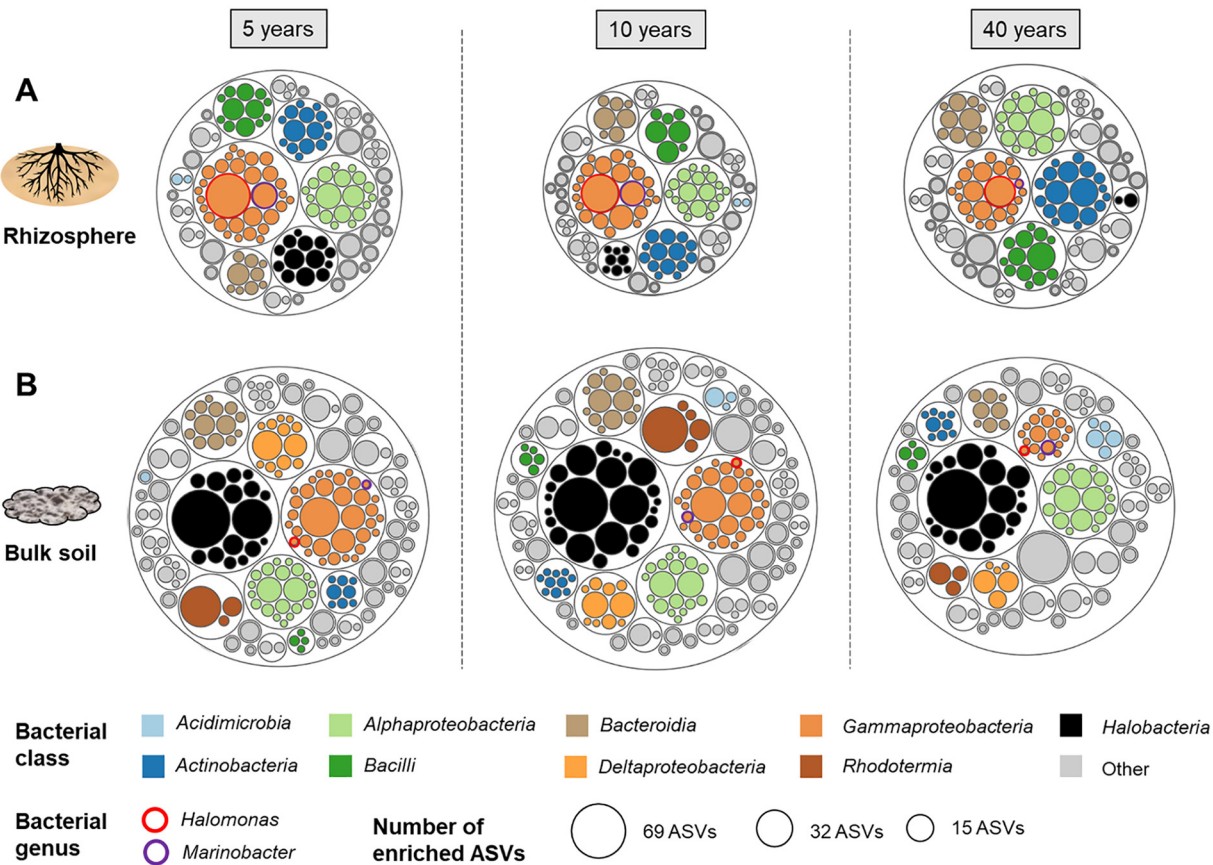

**FIG 2** Numbers of bacterial and archaeal ASVs that were enriched in the rhizosphere (A) in comparison to bulk soil (B) in the same area. The assessment is based on amplicon sequencing data.

changes were observed. *Gammaproteobacteria* dominated both bulk soil and the rhizosphere (averages of 20.9% and 40.1%, respectively) (Fig. 1C), but then their relative abundances decreased along the desiccation gradient. Simultaneously, in bulk soil and rhizosphere, the relative abundance of *Actinobacteria* increased, with values of 0.8, 1.9, and 7.5%, along the desiccation gradient (bulk soil, 0.8, 1.9, and 7.5%; rhizosphere, 6.2, 8.4, and 28.9%). A similar trend was also observed with two other bacterial taxa, i.e., *Bacilli* and *Alphaproteobacteria*. In contrast, two dominant archaeal taxa, *Halomicrobiaceae* and *Haloferacaceae*, showed a decrease in their relative abundances along the desiccation gradient (*Halomicrobiaceae*, 38.0, 33.9, and 26.4%; *Haloferacaceae*, 48.4, 24.6, and 25.4%) (Fig. 1D).

On the basis of the above-mentioned results, we then focused on bacterial and archaeal taxa enriched in the rhizosphere in comparison to surrounding soil from the same area, because they might support the plants in order to resist the drought and salt stress. A high number of bacterial amplicon sequence variants (ASVs) (45 of 365 ASVs) that were enriched in the rhizosphere from the areas that were dried out for 5 years were identified as *Halomonas* and *Marinobacter* (*Gammaproteobacteria*) (Fig. 2). We hypothesized that these taxa play an important role in the early revegetation event. A total of 378 ASVs were enriched in the rhizosphere from the areas that were dried out for 40 years in comparison to surrounding soil from the same area. Interestingly, 58% (221 ASVs) of these ASVs could not be found in bulk soils from the areas that were dried out for 5, 10 and 40 years. Furthermore, ASVs that belong to *Nesterenkonia*, *Glycomyces* (*Actinobacteria*), *Planococcus* (*Bacilli*), and *Sphingomonas* (*Alphaproteobacteria*) emerged as bacterial genera that were enriched in the rhizosphere from the areas that were dried out for 40 years in comparison to surrounding

soil from the same area. With respect to the archaeal data set, 44 ASV were enriched in the rhizosphere in areas that were dried out for 5 years compared to the soil compartment from the same area. The majority of these ASVs were assigned to the genera *Haloterrigena*, *Natronoarchaeum*, and *Halomicrobium*. In comparison to the areas that were dried out for 5 years, only 9 and 12 archaeal ASVs were enriched in rhizosphere in comparison to the bulk soil from the areas that were dried out for 10 and 40 years, respectively (Fig. 2). The majority of these ASV belonged to the genus *Haloterrigena*.

**Rhizosphere samples harbored functionally adapted bacterial and archaeal communities which might support plant growth.** From the rhizosphere data set, beta diversity analysis based on KEGG orthology (KO) indicated bacterial and archaeal clustering according to the desiccation gradient (bacteria, $R^2 = 65.0\%$ and $P = 0.005$; archaea, $R^2 = 48.6\%$ and $P = 0.007$). We observed a clear clustering between soil and rhizosphere for bacterial functioning, whereas a lower degree of clustering was observed for archaeal functioning (Fig. S5C and D). Samples from areas that dried out 5 and 10 years ago showed a tendency to cluster closer together, whereas samples from the areas that dried out 40 years ago clustered separately. To identify specific differences in bacterial and archaeal functioning during the revegetation event, we focused on pairwise comparisons between areas that dried out 5 and 10 years or 40 years ago. Here, bioinformatic assessments using edgeR revealed that 3,486 KO categories (adjusted $P$ value [$P_{adj}$] < 0.1) (Fig. 3A) were significantly different between the sample groups. We identified a total of 488 bacterial genes and 152 archaeal genes with an abundance that was significantly different when areas that dried out 5 and 10 years ago were compared (Fig. 3A and B). The number of differentially abundant genes was substantially higher, with 2,535 bacterial genes and 311 archaeal genes when areas that dried out 5 and 40 years ago were compared.

Subsequent analyses focused on bacterial and archaeal genes that make it possible for plants to thrive under desiccation and as such are potentially beneficial for the host plant under this harsh environmental condition. Genes encoding enzymes with EC numbers and transporter proteins were mainly affected by the desiccation gradient (Fig. 3A). We detected genes that are likely involved in drought, salt, and toxic-compound tolerance across the nine plant-associated metagenomes (Fig. 3C and D), i.e., spermidine/putrescine transport, osmoprotectant transport system permease, glycine betaine transporter, gluconate/$H^+$ symporter, and multisubunit $Na^+/H^+$ antiporter. Interestingly, we observed that the abundances of genes encoding gluconate/$H^+$ symporter (*gntP*) and multisubunit $Na^+/H^+$ antiporter (*mnhE*) that belong to *Archaea* were more abundant in the area that dried out 5 years ago than other areas (Fig. 3D). In contrast, abundances of analogous genes that belong to *Bacteria* were higher in the areas that dried out 10 and 40 years ago than the area that dried out 5 years ago (Fig. 3C).

Other detected functions, such as sugar and nitrogen metabolism, indicate interactions with the plant host. We observed significantly higher abundances of bacterial and archaeal genes involved in denitrification, i.e., *narG* and *narH*, in the area that dried out 5 years ago than the area that dried out 40 years ago (Fig. 3C and D). Interestingly, archaeal genes encoding sugar transporters were more prevalent during the early revegetation event (5 years ago), whereas abundances of analogous genes that belong to *Bacteria* were enriched during the later revegetation events (10 or 40 years ago). Multiple genes encoding enzymes that are required to convert toxic compounds, i.e., mercury (*merA*) and arsenic (*arsC*), to less harmful compounds or for their removal via molecular pumps for heavy metals (*czcB* and *arsB*) were enriched in the area that that dried out 40 years ago in comparison to the area that dried out 5 years ago. Moreover, multidrug resistance genes, i.e., *blt*, *marC*, and *mdeA*, which are involved in resistance to toxic compounds, were enriched in the area that dried out 40 years ago in comparison to the area that dried out 5 ago. It can be concluded that during the revegetation event, the shifts in microbial community structure were also followed by shifts in microbial functioning.

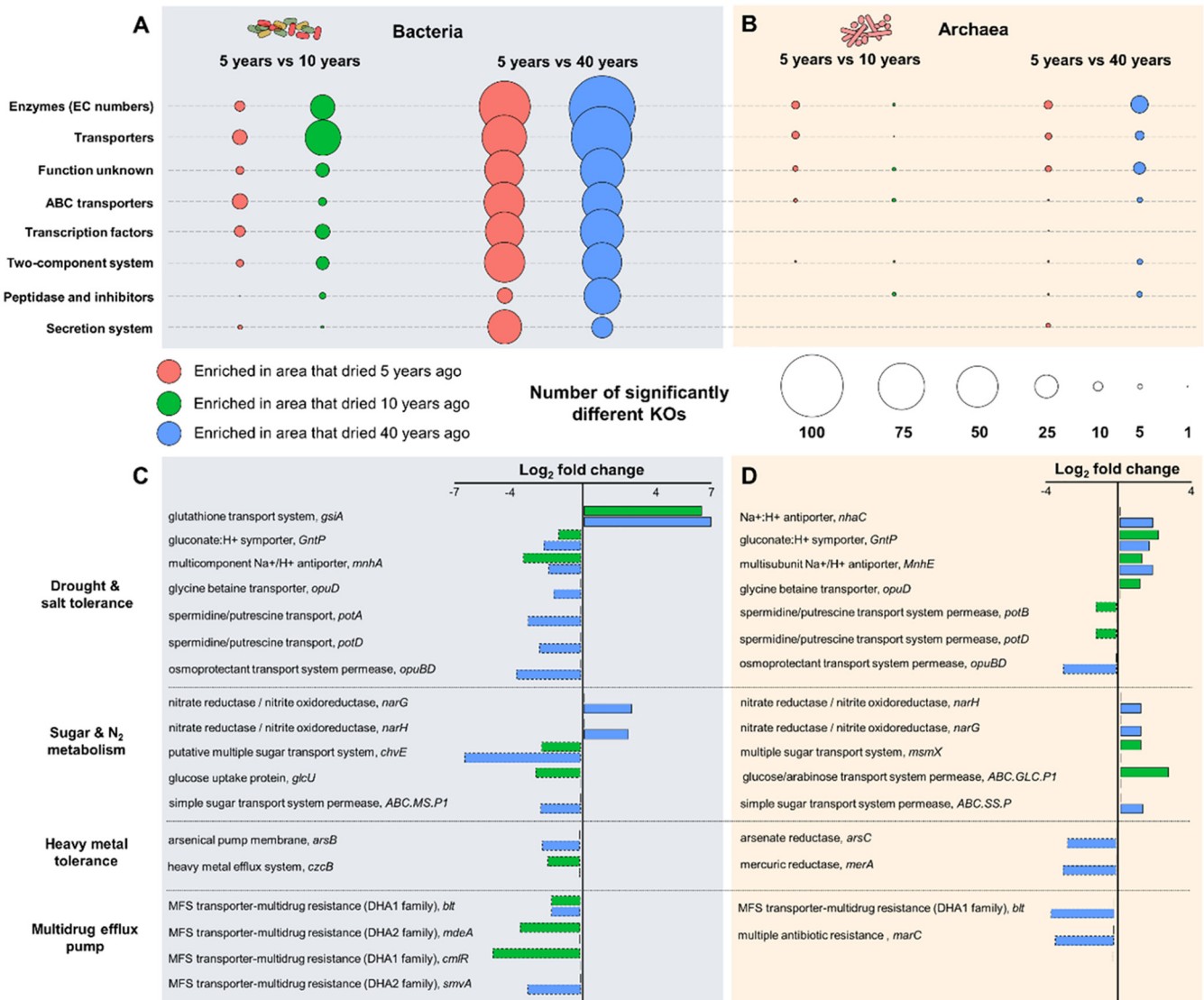

**FIG 3** Comparison of functional assignments based on KO categories that were either enriched or depleted within the desertification gradient within the rhizosphere community. The annotated bacterial (A) and archaeal (B) genes were assigned to their corresponding KO functional categories. The numbers of KO categories (A and B) that were either enriched or depleted according to differential abundance analysis are based on edgeR (log$_2$ fold change > 1; $P_{adj}$ < 0.1). The areas that were exposed to desiccation for 5 and 10 years (C) were separately assessed as well as the areas that were exposed to desiccation for 5 and 40 years (D). Red bubbles indicate the numbers of KO categories that were enriched in the areas that dried out 5 years ago compared to the areas that dried out 10 or 40 years ago. Green bubbles indicate KO categories enriched in the areas that dried out 10 years ago. Blue bubbles indicate KO categories that were enriched in the areas that dried out 40 years ago. Positive log$_2$ fold change values indicate that the respective bacterial (C) or archaeal (D) genes are more abundant in the areas that dried out 5 years ago than in areas that dried out 10 years (green) or 40 years (blue) ago, whereas positive values indicate that the genes are more abundant in areas that dried out 10 years (green) or 40 years (blue) ago than in areas that dried out 5 years ago.

**Key features related to metabolic adaptation and nutrient acquisition that can support host plants in bacterial and archaeal MAGs.** A total of 112 metagenome-assembled genomes (MAGs) with a completeness above 75% and contamination levels lower than 10% were recovered from the metagenomic data (Table S4). A majority of MAGs were identified as *Gammaproteobacteria* (n = 35), *Actinomycetia* (n = 19), *Halobacteria* (n = 13), and *Alphaproteobacteria* (n = 12). Of all high-quality bacterial MAGs (completeness of >90% and contamination of <5%), two were assignable only at the order level, while eight were assignable only at the family level (Fig. 4A; Table S4). Moreover, one archaeal MAG could not be assigned at the genus level despite a high genome completeness (Table S4), which additionally indicates a possible occurrence of novel bacterial and archaeal lineages in the dried-out Aral Sea basin (Fig. 4A).

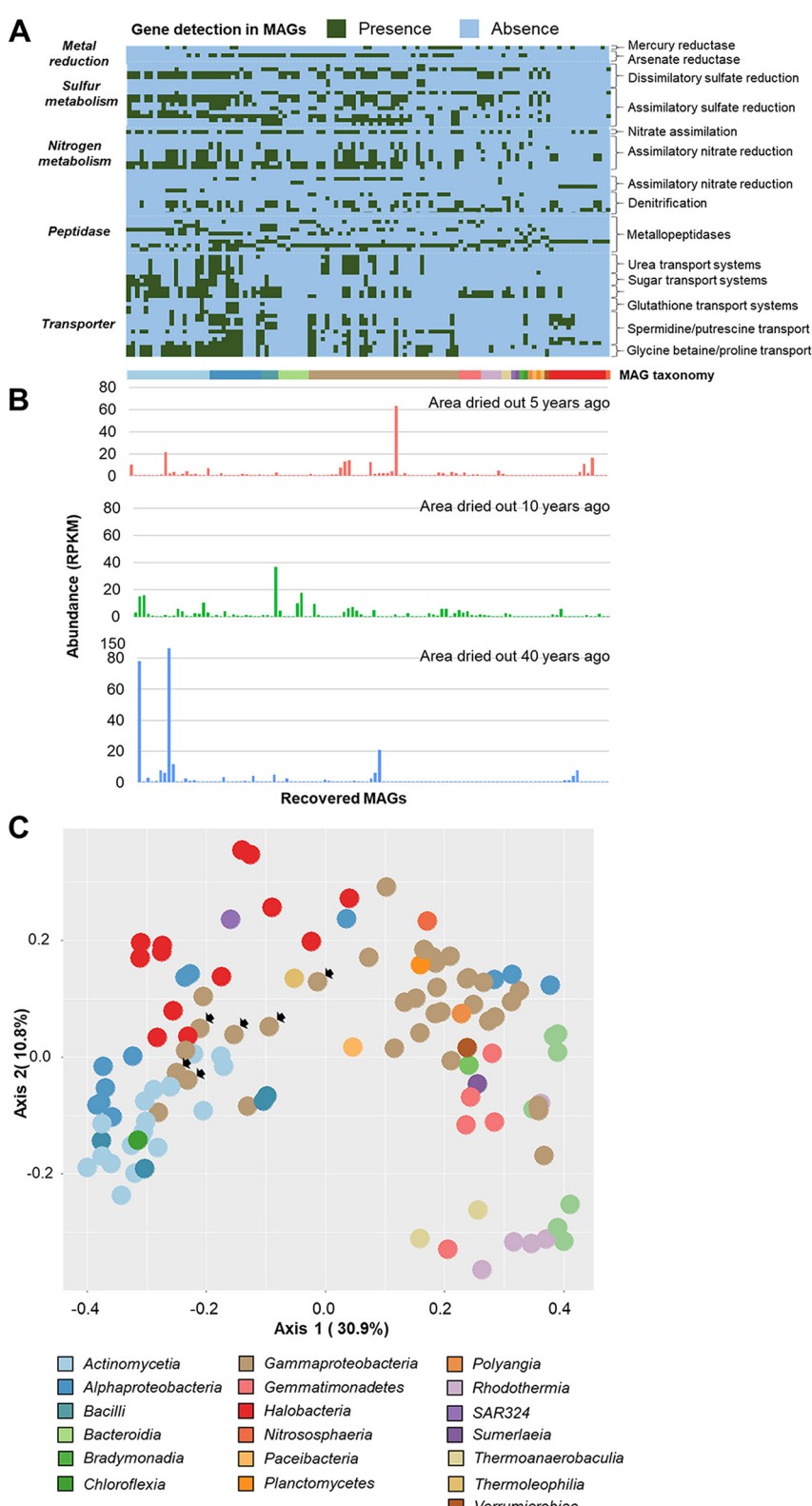

**FIG 4** Gene profiles for selected functions in MAGs recovered from the dried-out Aral Sea basin and clustering based on genes encoding transporter proteins from each MAG. Presence (green)/absence (blue) plots showing profiles of selected genes in MAGs recovered from the dried-out Aral Sea basin (A). Abundances of MAGs based on the number of mapped reads per kilobase per million reads (RPKM) (*n* = 3 replicates) in the 5-years-dry area (red), 10-years-dry area (green), or 40-years-dry area (blue) (B; detailed in Table S4). PCoA of MAG clusters is based on counts of genes encoding transporter proteins in the genomes (C). MAGs belonging to *Halomonas* are indicated with arrows (C).

Genome-centric analyses with representative bacterial and archaeal MAGs facilitated detailed assessments of their adaptability, their putative ecological functions, and implications for their host plant (Fig. 4A). A total of 47 MAGs harbored genes encoding nitrite reductase (*nir*) which converts nitrite to nitric oxide in the denitrification pathway. A majority of them belonged to *Pseudomonadales* ($n = 8$) and *Halobacteriales* ($n = 8$). Moreover, a total of 18 MAGs harbored a specific set of genes i.e., *narGHI* and *nirBD*, which is involved in dissimilatory nitrate reduction. The majority of them belonged to *Pseudomonadales* ($n = 8$) and were therefore assigned to the bacterial class *Actinomycetia* ($n = 8$), including *Actinomycetales*, *Mycobacteriales*, *Jiangellales*, *Propionibacteriales*, and *Streptosporangiales*. Only one MAG belonging to *Nitrososphaeria* harbored genes encoding a methane/ammonia monooxygenase. Two MAGs belonging to *Actinomycetales* harbored a set of genes involved in assimilatory sulfate reduction, which converts sulfate to hydrogen sulfide.

We detected high occurrences of transporter genes likely related to the adaptability of bacteria and archaea in the dried-out Aral Sea basin (Fig. 4A). Genes involved in spermidine (*potB*, *potC*, and *potD*), glycine betaine/proline (*proV*, *proW*, and *proX*), and osmoprotectant (*ABC.BCP1.S*, *ABC.BCP1.P*, and *ABC.BCP1.A*) transport systems were detected with high occurrences, particularly from MAGs belonging to *Actinomycetales* and *Mycobacteriales* (class *Actinomycetia*), *Rhizobiales* and *Rhodobacterales* (class *Alphaproteobacteria*), and *Halobacteriales* (class *Halobacteria*). Moreover, we also detected a total of 50 MAGs carrying *arsC*, a gene involved in resistance to arsenate, and a total of 24 MAGs carrying *merA*, a gene involved in resistance to mercury.

Additionally, the constructed principal-coordinate analysis (PCoA) plot indicated the occurrence of distinct clusters based on genes encoding transporter proteins (Fig. 4C), despite the fact that some of these MAGs were phylogenetically not closely related (Fig. S6). The majority of MAGs belonging to *Actinomycetia*, *Alphaproteobacteria*, and *Halobacteria* formed a separate cluster, whereas the majority of *Gammaproteobacteria*, except the genus *Halomonas*, formed another cluster together with *Gemmatimonadetes* and *Bacteroidia*. Genes encoding the glucose/mannose transport system (i.e., *gtsA*, *gtsB*, and *gtsC*) and the thiamine transport system (i.e., *ABC.VB1.P* and *ABC.VB1.A*) were more prevalent in MAGs that belong to *Halobacteria* (9 of 13 MAGs and 11 of 13 MAGs, respectively) and *Alphaproteobacteria* (6 of 12 MAGs and 7 of 12 MAGs, respectively). MAGs that belong to *Actinobacteria* (12 of 19 MAGs) and *Alphaproteobacteria* (6 of 12 MAGs) also harbored genes encoding multiple sugar transport systems (i.e., *ABC.GGU.A*, *ABC.GGU.S*, and *ABC.GGU.P*), indicating that their roles are to scavenge or access simple sugars and vitamin B1 from the surrounding environment. The biotin transport system substrate-specific component (*bioY*) protein was detected in a major proportion of these taxa (13 of 19 *Actinomycetia* MAGs, 7 of 12 *Alphaproteobacteria* MAGs, and 12 of 13 *Halobacteria* MAGs) but was mostly lacking in the other predominant taxonomic group, i.e., *Gammaproteobacteria* (4 of 35 MAGs). In contrast, genes encoding the phospholipid transport system (i.e., *mlaB* and *mlaC*) were highly prevalent in the latter taxonomic group (22 of 35 MAGs). Therefore, despite common features which indicate adaptation to the dried-out Aral Sea basin, distinct functional groups are present within naturally occurring bacterial and archaeal communities that were characterized by a high diversity of genes encoding transporter proteins.

To test whether the prevalence of specific MAGs could be linked to certain traits during the revegetation event, we examined MAGs that were differentially abundant between the early (areas that dried out 5 years ago) and late (areas that dried out 40 years ago) revegetation events. A total of 80 MAGs were differentially abundant between the two locations ($P_{adj} = 0.1$), where 61 of the total MAGs were enriched in areas that dried out 5 years ago. We found that the majority of MAGs belonging to *Gammaproteobacteria* and *Halobacteria* were enriched in this area (Fig. 4B; Table S4) whereas MAGs belonging to *Actinomycetia* and *Alphaproteobacteria* were enriched in the area that dried out 40 years ago. This finding indicates a shift from *Halobacteria* (*Archaea*) and *Gammaproteobacteria* in the early revegetation event to *Actinomycetia* and *Alphaproteobacteria* in the late revegetation event.

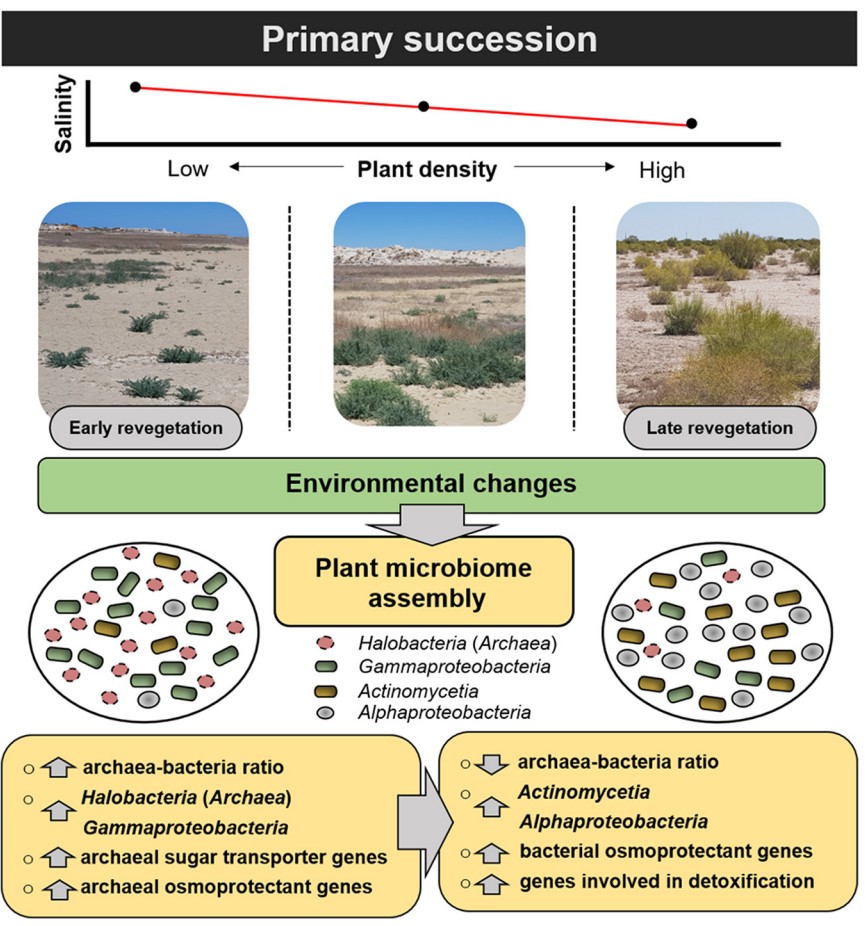

**FIG 5** Schematic illustration of microbiome adaptations within the desiccation gradient. Host plants enrich specific taxa with specific functions to dynamically adapt to prevalent environmental changes.

## DISCUSSION

The natural lab constituted by the dried-out Aral Sea basin allowed us to study rhizosphere assembly of the indigenous pioneer plant *Suaeda acuminata* across a time-dependent gradient of desiccation, which is characterized by decreased salinity and higher vegetation density (3). Here, we showed that microbiome structure and functioning also followed that gradient. While the rhizosphere effect could be confirmed in general, under these extreme conditions, we identified a novel, function-based mechanism for rhizosphere assembly. This was characterized by a replacement of archaeal taxa by bacteria across that gradient and shown in the decreasing archaeon-bacterium ratio and functional adaptation. This shows the high flexibility of pioneer plants, which first acquire archaea as supporters, which then, according to the salt and toxin gradient, get replaced by bacteria (summarized in Fig. 5).

We found significantly higher abundances but lower diversity of prokaryotes in the rhizosphere, indicating selective filtering and enrichment of a specific subset from the surrounding soil. This is characteristic of the well-known rhizosphere effect for bacteria (35), which is sparsely explored and not well understood for archaea (36, 37). The enrichment included specific bacterial taxa, i.e., *Halobacteria*, *Halomonas*, and *Marinobacter*, that were previously reported to be beneficial for plants in terms of increasing their tolerance to drought and salt stress (36, 38, 39). These taxa likely originated from the hypersaline body of water in the Aral Sea and were recruited to support plant growth specifically under high-salinity conditions. Interestingly, these taxa were replaced by *Actinomycetia* and *Alphaproteobacteria* under lower salinity, which are more typical rhizosphere inhabitants and commonly occur in the

plant-associated microbiota (35). Bergna and colleagues previously provided evidence that plant beneficial endophytes can be transmitted via seeds (30). We argue that *Actinomycetia* and *Alphaproteobacteria* originated from the host plant as endophytes or epiphytes and were then enriched over the time due to their beneficial roles; however, under extreme conditions, as found in regions that recently dried out, halophilic archaea prevail because they are better adapted to this environment. These results are in line with a recent study by Hanusch and colleagues (40) suggesting that more deterministic processes such as niche filtering and biotic interactions are the main drivers of community assembly over the course of succession. A recent study conducted in a hyperarid environment also suggested that plants may have coevolved with specific bacterial taxa that possess the capacity to improve the fitness and survival of the host plant (41). Our study suggests a strong host-driven selection for specific functional traits of the plant-associated microbial community over the course of the revegetation event. This contributes to fundamental understanding and managing the rhizosphere effect, which is crucial for plant production as well as revegetation strategies.

We observed a high abundance of genes associated with osmoprotectant transporters that increased along the desiccation gradient. Enrichments of such genes have been previously described as distinctive microbial responses to drought and salinity (42–44). Various bacterial and archaeal genes involved in detoxification of mercury and arsenic (i.e., *arsC* and *merA*) which were previously identified in similar settings (45–47) were found to be enriched along the desiccation gradient. This was also found for multidrug resistance genes, which followed the same pattern. The enrichment of these genes is likely due to the presence of various toxic agrochemical residues in the Aral Sea which accumulated in the dried-out sea bed (48).

Overall, a key role of archaea during the early revegetation stage was apparent in the present study. We observed a high abundance of archaeal genes that encode sodium/proton antiporters during this stage. Sodium/proton antiporters are ubiquitous in bacteria and also in archaea (49) and are required to regulate their pH and $Na^+$ homeostasis (50). They are also essential when microbes are exposed to salt stress in order to excrete $Na^+$ from the cells (49, 51, 52). We found that archaeal genes related to gluconate transporters (*gntP*) were also enriched during the early revegetation stage, whereas analogue bacterial genes were enriched during the later stages. Gluconate is one key metabolite involved in drought tolerance in various plants (53, 54). A recent study showed that application of external gluconate enhanced drought tolerance in *Oryza sativa* by increasing the root water uptake (55). Hence, it can be assumed that enrichment of archaeal taxa with these specific transporters can provide the required gluconate to mitigate salt stress of the host plant. The identified functions of archaeal and bacterial communities within the desiccation gradient indicate that plants can dynamically modify their microbiota to adapt to extreme environmental conditions (summarized in Fig. 5).

The reconstruction of MAGs allowed us to dissect functional implications of the microbiome with high taxonomic resolution and to identify key players within the desiccation gradient. These complementary analyses provided further evidence that bacterial as well as archaeal members were functionally adapted to drought and heavy metal stress in the Aral Sea basin due to the high occurrence of genes involved in osmoprotectant transport systems and heavy metal reduction. In addition, the analyses showed that *Gammaproteobacteria*, *Actinomycetia*, *Halobacteria*, and *Nitrososphaeria* are involved in denitrification, dissimilatory nitrate reduction to ammonium (DNRA) and ammonia oxidation. Denitrification is also known to play a major role in the production of the greenhouse gas nitrous oxide ($N_2O$) (56). However, the intermediate products, i.e., hydroxylamine and nitric oxide (NO), are signal molecules for plant adaptability toward abiotic stresses, including drought and salt (57–59). It is important to note that the presence of these genes does not necessarily confirm that they are functionally expressed. Nevertheless, the reconstructed MAGs can be used in the future for in-depth analyses by implementing stable isotope experiments coupled with RNA-based (metatranscriptomic) or protein-based (metaproteomic) approaches.

In the present study, we could also identify functional guilds that are based on genes encoding transporter proteins despite many of the analyzed MAGs not being closely related in terms of phylogeny. Clustering based on functional profiles rather than close phylogenetic relationships was previously described for metabolic specialization profiles (60, 61) but otherwise not commonly found (15, 62, 63). Our data indicate that microbiome assembly in plants that are commonly found in the dried-out Aral Sea basin is mainly based on the presence of specific genes rather than particular taxonomic groups. Therefore, we argue that in addition to the plant's innate ability to tolerate drought and salt stress, its ability to modulate microbiome assembly by cofiltering specific taxa with specific traits could be one mechanism by which plants survive under extreme conditions. By identifying key players of this process, we could design synthetic microbial communities for a desired plant phenotype in the future (64). However, the plant microbiome does not consist only of prokaryotic communities; eukaryotic microorganisms and viruses also colonize the rhizosphere and have to be integrated into those strategies (65). Viral populations in this habitat were studied in parallel; they are mainly driven by the gradient of desiccation and corresponding salinity as well as the rhizosphere bacterial populations (W. A. Wicaksono, D. Egamberdieva, T. Cernava, and G. Berg, submitted for publication). First insights into fungal communities showed that they are less responsive to the gradient but have to be studied in more detail (W. A. Wicaksono, unpublished data).

In conclusion, the desiccation gradient of the Aral Sea basin facilitated the establishment of bacterial and archaeal communities with functional traits that can support not only their own but also plant survival under extreme conditions. The enrichment of bacterial and archaeal taxa with supporting functions indicates a dynamic coadaptation with their host plants. This study also provides further evidence for the hypothesis that archaea, a commonly neglected component of the microbiome, play an important role to support pioneer plants under extreme conditions. Our results showed a function-driven rhizosphere assembly and support of plant growth by the microbiome. This has implications for the selection of microbial inoculants, i.e., synthetic bacterial communities to protect plants against biotic and abiotic factors, for the ongoing environmental restoration of the Aral Sea basin (Aralkum desert), and for the increasing number of global salt-affected soils (66).

## MATERIALS AND METHODS

**Sample collection and processing.** The South Aral Sea belongs to Uzbekistan (45°00′ N, 60°00′ E). We collected bulk soil and rhizosphere samples of the plant *Suaeda acuminata* (C.A.Mey.) Moq. in the dried-out basin and near the west shoreline of the South Aral Sea. The samples were obtained from areas that dried out 5 years, 10 years, and 40 years ago. From each area, samples were taken from three independent biological replicates (from three different *S. acuminata* populations), each consisting of combined roots of three individual plants to obtain at least 5 g material per biological replicate. Rhizosphere samples were collected by light shaking of the roots to remove loosely attached soil before they were further treated in the laboratory as described below. These sampling sites were previously studied with a comprehensive experimental design and metadata in terms of geochemistry and mineralogy (3). Soil contained sand, clay, and silt, at 37.9%, 53.2%, and 8.9%, respectively. Salinity (total soluble salt) decreased from 67.1 g/L in the area that dried out 5 years ago to 0.4 g/L in the one that dried out 40 years ago (Fig. 1A). The salinity was negatively correlated with the number of occurring plant species in the studied area (Fig. S1). From these locations, collected soil and rhizosphere samples were placed in sterile bags and kept in a cooling box before being transferred to our laboratory facility for processing and storage under controlled conditions.

To isolate the total DNA from bulk soil and the rhizosphere samples, either bulk soil or plant roots with adhering rhizosphere soil were mixed with 20 mL sterile 0.85% NaCl and subsequently vortexed for 3 min. Fractions of 2 mL of the resulting suspensions were centrifuged at 16,000 $\times$ *g* and 4°C with a Sorvall RC-5B refrigerated superspeed centrifuge (DuPont Instruments, USA) for 20 min. The pellets were weighed (approximately 0.1 g) and stored at $-$20°C until DNA extraction using the FastDNA spin kit for soil (MP Biomedicals, USA) following the manufacturer's protocol.

**Quantitative real-time PCR.** Total DNA was further used for qPCR with the primer set 515f/806r to amplify the V4 region of prokaryotic 16S rRNA genes with the PCR parameters described previously (67). For archaeal specific qPCR, we used archaeon-specific primers 344aF and 517uR (68, 69). The quantification was performed with a Rotor-Gene 6000 thermal cycler (Corbett Research, United Kingdom) and SYBR green PCR master mix (KAPA Biosystems, USA). All PCRs were performed in two technical replicates. Serial dilutions of a standard containing a defined number of 16S rRNA gene copies from *Bacillus* sp. and *Haloferax denitrificans* were used for the calculation of bacterial and archaeal gene copy

numbers, respectively, in different samples. The copy numbers of bacterial and archaeal 16S rRNA genes were normalized to the weight of the samples. To obtain the bacterial abundance, we subtracted archaeal abundance from prokaryotic abundance. Statistical analysis of the bacterial and archaeal abundance was conducted using the Kruskal-Wallis test followed by a pairwise Dunn test.

**Amplicon sequencing of prokaryotic markers genes and shotgun metagenomic sequencing of total community DNA.** The primer set 515f/806r was used to amplify the V4 region of prokaryotic 16S rRNA genes with the PCR parameters described previously (67). The PCR mixture (25 $\mu$L) contained 1$\times$ Taq&Go (MP Biomedicals, Illkirch, France), a 0.25 mM concentration of each primer, and 1 $\mu$L template DNA. Barcoded PCR products were purified using the Wizard SV gel and PCR clean-up kit (Promega), pooled in equimolar concentrations, and then sequenced using an Illumina MiSeq PE 300 instrument by the sequencing provider Genewiz (Leipzig, Germany). Moreover, shotgun metagenomic sequencing was performed on total DNA extracted from bulk soil and rhizosphere samples using an Illumina HiSeq PE 150 instrument. Due to low DNA concentrations in the bulk soil samples, the triplicates had to be pooled for each soil location prior to library construction and shotgun metagenomic sequencing.

**Processing of raw sequencing data.** QIIME2 version 2019.10 (https://qiime2.org) (70) was implemented to analyze the amplicon sequencing data set. Raw reads were demultiplexed with the cutadapt tool (71). Primer sequences were trimmed using the cutadapt plugin (71) and then subjected to quality filtering, denoising, and chimeric sequence removal using the DADA2 algorithm (72). The resulting amplicon sequences variants (ASVs) were subsequently aligned against the reference database Silva v132 (73) using the VSEARCH classifier (74) to obtain taxonomical information of each ASV. Prior to further analyses, only reads assigned to *Bacteria* and *Archaea* were retained. Amplicon sequencing resulted in a total of 951,675 bacterial reads (average, 52,870; range, 14,206 to 86,529) and 234,734 archaeal reads (average, 13,040; range, 276 to 34,893) (Table S1) which were assigned to a total of 7,302 bacterial and 1,772 archaeal ASVs.

For the shotgun metagenomic data set, Kraken2 was used to classify individual metagenomic reads by mapping all k-mers to the lowest common ancestor (LCA) of all reference genomes (75). Estimations of species abundances were conducted with Bracken (76). Shotgun-based metagenomic sequencing resulted in a mean number of 45,300,754 high-quality reads per sample (range, 34,001,522 to 56,395,621) (Table S1). Using the Kraken2 classifier, a total of 1,407 bacterial and 121 archaeal genera were identified in the metagenome library. Subsequently, for alpha diversity analysis, the species abundance table was normalized to the lowest number of reads to account for uneven sequencing depth. MetagenomeSeq's cumulative sum scaling (CSS) (77) was used for subsequent beta diversity analyses.

The microbial data sets were analyzed with the R package Phyloseq implemented in RStudio (78, 79). The Shannon diversity index was used to estimate alpha diversity and a nonparametric Kruskal-Wallis test to determine the significance of observed differences. Beta diversity analysis was performed using a normalized Bray-Curtis dissimilarity matrix. The matrix was subjected to analysis of similarities (ANOSIM) to test for significant effects in the microhabitat and desertification gradient. Finally, the DESeq2 method (80) was used to determine the differences between the ASV abundances in different microhabitats and within the desiccation gradient. Prior to using the DESeq2 method, the number of reads was normalized to the gene copy number obtained with the qPCR approach in order to obtain a more reliable differential abundance among samples. ASVs were considered significantly different if the adjusted $P$ value (Benjamini-Hochberg adjustment) was less than 0.1 and the log fold change was >2.

**Assembly-based metagenomic analyses.** Trimmomatic and VSEARCH were implemented to remove Illumina sequencing adaptors and to perform initial quality filtering. Metagenomic reads were then assembled using MEGAHIT (81). Only contigs with a length of >1 kb were retained for subsequent analyses. Open reading frames were predicted using Prodigal (82). To remove redundant sequences, CD-HIT-EST was used; it clusters protein-coding gene sequences into a nonredundant gene catalogue using a nucleotide identity of 95% similarity (83). Nonredundant genes were annotated using the BLAST algorithm in DIAMOND in combination with eggNOG-mapper (84, 85) and the eggNOG database (86).

To explore ecologically relevant functions, only contigs with retrievable KEGG orthology (KO) annotation information were further analyzed. Quality-filtered reads were back-mapped to the specifically generated nonredundant gene catalogue using Burrows-Wheeler aligner (BWA) and SAMtools (87, 88) to obtain abundances for specific genes. Within the metagenomic data sets, totals of 18,679,144 reads (range, 124,704 to 3,490,074) (Table S1) and 248,031,666 reads (range, 10,783,281 to 32,428,407) were classified as archaeal and bacterial proteins according to eggNOG-mapper. The output was subjected to analyses with edgeR to identify differentially abundant genes in different sample types. Beta diversity analysis to test for significance in KO composition was performed as described above.

**Reconstruction of bacterial and archaeal metagenome-assembled genomes.** Maxbin2, MetaBAT2, and CONCOCT (89–91) were used to reconstruct MAGs from bulk soil and rhizosphere samples. To dereplicate the MAGs, DASTool (92) was implemented. The quality of MAGs was calculated using CheckM (93). Only medium-quality MAGs, as determined according to the current definition of the minimum information metagenome-assembled genome (MIMAG) standards (94), with at least 75% completeness were kept for further analyses. Metagenome-assembled genomes were then dereplicated using dRep v2.2.3 (95) to obtain a nonredundant metagenome-assembled bacterial genome set. GTDB-Tk (96) was utilized to obtain taxonomical information for each MAG. Phylogenetic trees were constructed using PhyloPhlAn (97). Prediction of protein-coding sequences and gene annotations were performed using DRAM (98). Abundance profiles of each MAG were estimated by using CoverM with the option -m rpkm, –min-read-aligned-percent 0.75 and –min-covered-fraction 0. Differences in MAG abundances within the desertification gradient were analyzed using a pairwise Wilcoxon test.

**Data availability.** This shotgun metagenome project has been deposited in the European Nucleotide Archive (ENA) database under the numbers PRJEB51330 (amplicon sequencing data set) and PRJEB51329 (shotgun metagenome data set).

## SUPPLEMENTAL MATERIAL

Supplemental material is available online only.

**FIG S1**, TIF file, 2.8 MB.
**FIG S2**, TIF file, 0.7 MB.
**FIG S3**, TIF file, 0.5 MB.
**FIG S4**, TIF file, 0.2 MB.
**FIG S5**, TIF file, 0.3 MB.
**FIG S6**, TIF file, 0.4 MB.
**TABLE S1**, DOCX file, 0.01 MB.
**TABLE S2**, DOCX file, 0.01 MB.
**TABLE S3**, DOCX file, 0.01 MB.
**TABLE S4**, DOCX file, 0.01 MB.

## ACKNOWLEDGMENTS

We thank Maged Saad (KAUST), Julia Kranyeck, and Kristina Michl (Graz) for their support during sampling, DNA extraction, sample preparation, and molecular work.

G.B., C.B., T.C., and D.E. designed the study and conducted the sampling. M.M. performed all molecular work. W.A.W. and P.K. analyzed the data. W.A.W., T.C., P.K., and G.B. interpreted the data and wrote the manuscript. All authors critically read the final draft.

We declare that we have no competing interests.

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
