## [Reviewer comments · mSystems]

Function-based rhizosphere assembly along a gradient of desiccation in the former Aral Sea

Wisnu Adi Wicaksono, Dilfuza Egamberdieva, Christian Berg, Maximilian Mora, Peter Kusstatscher, Tomislav Cernava, and Gabriele Berg

Corresponding Author(s): Tomislav Cernava, Graz University of Technology

Review Timeline:

Submission Date:	August 3, 2022
Editorial Decision:	September 19, 2022
Revision Received:	October 27, 2022
Accepted:	October 29, 2022

Editor: Karoline Faust

Reviewer(s): Disclosure of reviewer identity is with reference to reviewer comments included in decision letter(s). The following individuals involved in review of your submission have agreed to reveal their identity: Ramona Marasco (Reviewer #2)

Transaction Report:

DOI: <https://doi.org/10.1128/msystems.00739-22>

September 19, 2022

Dr. Tomislav Cernava
Graz University of Technology
Institute of Environmental Biotechnology
Graz
Austria

Re: mSystems00739-22 (Function-based rhizosphere assembly along a gradient of desiccation in the former Aral Sea)

Dear Dr. Tomislav Cernava:

Thank you for submitting your manuscript to mSystems. The reviewers have now assessed your work and recommend it for publication after a number of issues have been addressed. For instance, please check metagenomic data for eukaryotic reads to see whether fungal communities may play a role in this ecosystem. The reviewers also recommend deepening the functional analysis, taking biochemical data such as salinity into account. Such data could also help to quantify what is meant by "extreme" desiccation.

Below you will find instructions from the mSystems editorial office and comments generated during the review.

Preparing Revision Guidelines

Sincerely,

Karoline Faust

Editor, mSystems

Journals Department
Reviewer comments:

Reviewer #1 (Comments for the Author):

In the manuscript "Function-based rhizosphere assembly along a gradient of desiccation in the former Aral Sea", as the authors indicated, these results confirm that along the desiccation gradient, a significant change in the rhizosphere microbiota were observed in the rhizosphere microbiota of native pioneer plants. The results provide evidence that rhizosphere assembly by co-filtering specific taxa with distinct traits is a mechanism, which allows plants to thrive under extreme conditions. Here are some suggestions about this manuscript.

1. Clearly explain the significant changes in the rhizosphere microbiota of native pioneer plants across a gradient of desiccation (5, 10, and 40 years) were obtained in the abstract.
2. Line 32-34, "The rhizosphere effect was shown by significantly higher abundances but less diversity in the rhizosphere compared to bulk soil in all samples." Do bacterial and archaeal communities have the same alteration trend?
3. Why bacterial and archaeal communities were investigated rather than fungal communities?
4. Line 113-115, "Although *S. acuminata* is not a classical pioneer plant, it fulfils this role under the specific extreme conditions in the Aral Sea basin.". Reference should be cited.
5. Line 126-128, "We collected bulk soil and rhizosphere samples (three biological replicates - approx. 5 g per biological replicate) of the plant *Suaeda acuminata* (C.A.Mey.) Moq. in the dried-out basin and near the west shoreline of the South Aral Sea". Overall, my concern is why *Suaeda acuminata* was select?
6. Line 296, delete an "in".
7. In legend of figure 1, (A), (B), (C) and (D) should be explained clearly.
8. In legends of figure 1 and 5, "High", "Medium" and "low" should be explained clearly.

Reviewer #2 (Comments for the Author):

The manuscript mSystems00739-22 by Wicaksono and coauthors reports an investigation of the compositional and functional diversity and quantification of prokaryotes (bacteria and archaea) associated with the rhizosphere of native pioneer halophytic *Suaeda acuminata* along a desiccation gradient in the former Aral Sea. Metagenome and amplicon sequences approach, along with qPCR, were applied to 12 samples (9 rhizospheres and 3 soils) across three sites exposed to desiccation for 5, 10 and 40 years. Different bacterial microbiomes were found in the rhizosphere and soil, with different structures and assemblies among the sites, due to the combination of plant co-filtering and environmental conditions of sites (desiccation, salt concentration, toxin, etc.). Notably, along the desiccation gradient, the authors showed that the rhizosphere microbial communities had a decreasing archaea-bacteria ratio, a replacement of halophilic archaea by certain bacteria, and an adaptation of specific beneficial microbial functions, such as carbon cycling and fixation, methane and nitrogen metabolism, and production of osmoprotectants, which could support plants to thrive under extreme conditions.

Overall, the manuscript is well written, the results are clearly presented, and the conclusions are well supported.

Below are some minor criticisms and my comments/suggestions:

Line 33. Which pioneer plant? Maybe it can be specified from the beginning. Are all the collected plants the same species? Is this evaluated?

Line 35. Why is the site with 5 years of desiccation considered the most extreme? Not really get this sentence. I suppose that at this site the salt concentration is the highest. I suggest reporting at least this info related to the three sites also in the abstract.

Line 42. Are these MAGs coming only from the rhizosphere?

Line 99-103. While Lin et al 2022 describe the overall trend of the rhizosphere microbiome in terms of composition/assembly and predicted function/trophic behaviours, a work recently published confirmed the predicted functionality by applying the metagenome approach (see 10.1038/s43705-022-00130-7). The authors showed that the root system represents a microbial density and competition hotspot ruled out by a dual selection process: to colonize the plant-associated niches microorganisms must (1) possess the capacity to improve the fitness and survival of the plant (plant selection of beneficial microbes, PGP bacteria) and be able (2) to compete successfully against other microorganisms (microbial competition). It will be interesting to explore if this competition is also present in salty/desiccation soils and how this occurs along the desiccation gradient.

Lines 108-109. Among others, 10.1038/s41396-022-01238-3 showed how the autochthonous community in desert soil promotes the growth of plants, suggesting the potential of edaphic microbes and their ability to further colonize new plants and exert their beneficial services. I suggest better discussing the functional aspect of soil community and interaction with plants.

Line 130. How did you collect 5 g of rhizosphere? How many plants were used for each replicate? Is this species the main abundance? How is it distributed along the desiccation gradient?

Line 131. How was plant species identified during the sampling? Did you perform a molecular check on all samples collected to confirm they are the same species? Are the plants selected in the same phenological stage and have the same size? Could the latest be affected by salinity/desiccation? This is also an interesting point to be explored/discussed/tested.

Lines 142-143. This passage is not clear. You added 20 ml of solution, but you stated that the resulting suspension was 2ml. Did you just collect a portion? Please clarify and add the range of soil used for DNA extraction (weighted after the centrifuge) so that

readers have an idea.

Lines 156-157. How were qPCR results normalized? Based on the taxonomy obtained the different copy numbers of 16S rRNA of bacterial and archaea communities could be retrieved across the three sites.

Line 179. Why Silva 128? A more recent version (138) is available. Is the classifier trained with the primer used? Please specify how many chloroplasts (and non-bacteria) were obtained and provide rarefaction curves. Same for metagenomes.

Line 202. Is the dataset also normalized based on the number of the 16S rRNA gene present in different taxa/ASVs?

Line 248 repetition of "dried out"

Line 240. The authors should briefly describe the condition of the three sites, and, if it is possible, also include C and other nutrient contents.

Lines 268-269. I suggest keeping the definition of the gradient as "desiccation" and not temporal. What do you mean by "Temporal changes"? If all the plants and soil were collected on the same day, the main changes observed in the rhizosphere/soil across the three sites are due to the "desiccation gradient". Moreover, the same desiccation is correlated/explained by time, so only one of these factors can be used to describe/explain the differences observed.

Line 306. Remove highlight from "Glycomyces".

Lines 327-333. Specify the type of genes included. Are all those differentially distributed across samples, or only those related to PGP/beneficial traits?

Line 436. Why do the mechanisms describe here can be defined as new? Several works showed how the rhizosphere tends to select core functional microbiome, as well as microbes enriched in PGP functions/traits that support plant growth. If this statement refers only to the archaea/bacteria ratio, please clarify it.

Figure 1. specify from amplicon dataset as done in Figure 2.

Figure 4A. maybe it can be clustered based on the distribution of metabolism across MAGs or on the distribution of MAGs in the three sites. It is difficult to follow the pattern described.

Figure 5. Correlation with chemistry data already published could additionally explain the pattern observed, especially trophic behaviours and C-metabolisms. Structural equation modelling (SEM) could be useful to explain the pattern observed, as well as it could be interesting to explore the correlation between microbes involved in different carbon and nitrogen cycling at different sites (e.g., pathway and corresponding percentage of genomes encoding the respective genes by using METABOLIC doi:10.1186/s40168-021-01213-8).

Table S1. What does star (*) means? From the caption, it is not clear. I suppose that pools were performed for bulk soil samples. However, from the methods, it is not clear if this was done for both amplicon and metagenome sequencing.

Tables S2 and S3. Please specify that these tables refer to the amplicon datasets. What does star (*) means? From the caption, it is not clear.

Table S5. I suppose this comes from the rhizosphere. Please specify. I also suggest, since three replicates were analyzed, adding the standard deviation to the relative abundance reported. If it is not possible, specify that the three metagenomes were pooled/merged to obtain MAGs; if it is the case how is relative abundance calculated? How did you compare? Which statistical analysis was applied?

Supplementary Figure S1. It could be interesting to present another set of panels with the same results from metagenomes. As well as add rarefaction curves for amplicon and metagenomes approaches, at least based on 16S rRNA diversity in the rhizosphere and soil across the three-time points.

Supplementary Figure S3. What was used to calculate Shannon diversity? Did you use the 16S rRNA genes, MAGs, or functional genes from the metagenome approach? Please specify. Moreover, why are shown four black dots per box plot if the rhizosphere samples are three per time-point? See NMDS and Table S1.

Supplementary Figure S4. Are A and C referring to the compositional diversity of community based on the 16S rRNA gene from metagenome/MAGs? Which taxonomic level was considered? Which functions were included in B and D? Please specify.

Supplementary Figure S5. From which compartment the MAGs showed were obtained? Rhizosphere and/or soil?

Dear Dr. Tomislav Cernava:

Thank you for submitting your manuscript to mSystems. The reviewers have now assessed your work and recommend it for publication after a number of issues have been addressed. For instance, please check metagenomic data for eukaryotic reads to see whether fungal communities may play a role in this ecosystem. The reviewers also recommend deepening the functional analysis, taking biochemical data such as salinity into account. Such data could also help to quantify what is meant by "extreme" desiccation.

Below you will find instructions from the mSystems editorial office and comments generated during the review.

Sincerely,

Karoline Faust

Editor, mSystems

→ Dear Prof. Faust,

Thank you for the opportunity to revise our manuscript titled "Function-based rhizosphere assembly along a gradient of desiccation in the former Aral Sea".

We have carefully considered all of the highly constructive comments provided by the reviewers and revised our manuscript accordingly. Thanks also for comments you pointed out. We analyzed metagenomic data for eukaryotic reads to see whether fungal communities may play a role in this ecosystem. Here, we have an answer – they seem to be less responsive to the parameter along the gradient. We decided to add this in the discussion section but not to the manuscript because this was not the scope of our study. Data are shown in the rebuttal letter. In addition, we have performed a detailed functional analysis, which was enclosed.

We appreciate all helpful comments and hope that the new version of our manuscript merits to be published in **mSystems**.

Reviewer #1 (Comments for the Author):

In the manuscript "Function-based rhizosphere assembly along a gradient of desiccation in the former Aral Sea", as the authors indicated, these results confirm that along the desiccation gradient, a significant change in the rhizosphere microbiota were observed in the rhizosphere microbiota of native pioneer plants. The results provide evidence that rhizosphere assembly by co-filtering specific taxa with distinct traits is a mechanism, which allows plants to thrive under extreme conditions. Here are some suggestions about this manuscript.

→ Thank you for your constructive comments. We have addressed your comments point-by-point as you will see below. They have certainly helped us to improve the manuscript.

1. Clearly explain the significant changes in the rhizosphere microbiota of native pioneer plants across a gradient of desiccation (5, 10, and 40 years) were obtained in the abstract.

→ The changes were reflected by differences in the archaea-bacteria ratio and the replacement of halophilic bacteria by typical plant-associated bacteria with specific plant-beneficial biosynthetic pathways. We revised the sentence in the Abstract in line 37 as follows:

"Along the desiccation gradient, we observed a significant change in the rhizosphere microbiota, which was reflected by i) a decreasing archaea-bacteria ratio, ii) replacement of halophilic archaea by specific plant-associated bacteria i.e., *Alphaproteobacteria* and *Actinobacteria*, and iii) an adaptation of specific, potentially plant-beneficial biosynthetic pathways."

2.Line 32-34, "The rhizosphere effect was shown by significantly higher abundances but less diversity in the rhizosphere compared to bulk soil in all samples." Do bacterial and archaeal communities have the same alteration trend?

→ Thank you for pointing this out. Yes, the rhizosphere effect in terms of the decrease of diversity, was consistently shown for the bacterial and archaeal communities (except for the archaeal abundance that is only higher in rhizosphere soil in comparison to bulk soil in the area that dried out 40 years ago).

3.Why bacterial and archaeal communities were investigated rather than fungal communities?

→ Indeed, the fungal community is an interesting topic. Currently there are only limited studies addressing microbial communities in the Aral Sea basin. We focused on bacteria due to their relatively high abundance in plant-associated habitats. We hypothesized that they would also play important roles in this extreme environment. Archaea that also commonly

colonize extreme environments and we therefore hypothesize that they might have implications for supporting host plants in this environment.

We performed additional analyses to investigate the impacts of the desiccation gradient on the fungal community structure based on shot-read metagenome sequencing. Interestingly, the fungal community structure was less responsive to the gradient of desiccation, but two major clusters were found ($R^2=37.9\%$, $P=0.069$, Figure 1). Due to the microbial dynamics in this environment, findings from this study provide a solid basis for larger, follow-up studies that include other members of the plant holobiont, e.g. fungi that were not in the scope of this study.

Figure 1. Fungal community structure in the rhizosphere of *Suaeda acuminata*.

4.Line 113-115, "Although *S. acuminata* is not a classical pioneer plant, it fulfils this role under the specific extreme conditions in the Aral Sea basin.". Reference should be cited.
→ We have included a reference in the text.

5.Line 126-128, "We collected bulk soil and rhizosphere samples (three biological replicates - approx. 5 g per biological replicate) of the plant *Suaeda acuminata* (C.A.Mey.) Moq. in the

dried-out basin and near the west shoreline of the South Aral Sea". Overall, my concern is why *Suaeda acuminata* was selected?

→ *Suaeda acuminata* was chosen because it is the first pioneer plant colonizing the dried-out Aral Sea basin, and the only one present along the gradient. Hence, this allowed us to compare its prokaryotic community structures along the natural revegetation stages.

6. Line 296, delete an "in".

→ We have deleted it.

7. In legend of figure 1, (A), (B), (C) and (D) should be explained clearly.

→ Thank you for the comment. We agree that the details are missing for the figure legend.

To avoid reader's confusion, we have revised the figure legend as follows:

Figure 1. Sampling site, bacterial and archaeal abundance, diversity and community structure in bulk soil and the rhizosphere of *Suaeda acuminata*. Different sampling sites represent the gradient of salinity (high, medium, and low salinity) and natural revegetation events in the Aral Sea basin where bulk soil and rhizosphere samples were collected (A). Geochemistry, mineralogy, and the number of visible plants species were previously described (Jiang et al., 2020). Bacterial and archaeal 16S rRNA gene copy numbers were calculated by using qPCR (B, C, and D). The diversity of bacterial (C) and archaeal (D) communities was estimated using the Shannon index in bulk and rhizosphere soils within the analyzed desiccation gradient (5 – 40 years).

8. In legends of figure 1 and 5, "High", "Medium" and "low" should be explained clearly.

→ Thank you for pointing this out. We have revised the figure and clearly explained the gradient of salinity in the figure legend.

Reviewer #2 (Comments for the Author):

The manuscript mSystems00739-22 by Wicaksono and coauthors reports an investigation of the compositional and functional diversity and quantification of prokaryotes (bacteria and archaea) associated with the rhizosphere of native pioneer halophytic *Suaeda acuminata* along a desiccation gradient in the former Aral Sea. Metagenome and amplicon sequences approach, along with qPCR, were applied to 12 samples (9 rhizospheres and 3 soils) across three sites exposed to desiccation for 5, 10 and 40 years. Different bacterial microbiomes were found in the rhizosphere and soil, with different structures and assemblies among the sites, due to the combination of plant co-filtering and environmental conditions of sites (desiccation, salt concentration, toxin, etc.). Notably, along the desiccation gradient, the authors showed that the rhizosphere microbial communities had a decreasing archaea-bacteria ratio, a replacement of halophilic archaea by certain bacteria, and an adaptation of specific beneficial microbial functions, such as carbon cycling and fixation, methane and

nitrogen metabolism, and production of osmoprotectants, which could support plants to thrive under extreme conditions.

Overall, the manuscript is well written, the results are clearly presented, and the conclusions are well supported.

→ Thank you for the positive comments. We appreciate the constructive suggestions. We have addressed all of your comments and have carefully edited the manuscript. We believe the revised version has improved.

Below are some minor criticisms and my comments/suggestions:

1. Line 33. Which pioneer plant? Maybe it can be specified from the beginning. Are all the collected plants the same species? Is this evaluated?

→ We agree that the name of plant needs to be specified in the beginning. We added this information in line 31 as follows:

“Here we investigated the rhizosphere microbiota of the native pioneer plant *Suaeda acuminata* (C.A.Mey.) Moq, in comparison to bulk soil across a gradient of desiccation (5, 10, and 40 years) by metagenome and amplicon sequencing combined with qPCR analyses.”

Yes, they are from the same species. We have identified the plant species according to its specific morphology.

2. Line 35. Why is the site with 5 years of desiccation considered the most extreme? Not really get this sentence. I suppose that at this site the salt concentration is the highest. I suggest reporting at least this info related to the three sites also in the abstract.

→ Yes, this is true. The area that dried out 5 years ago is considered the most extreme due to high salinity which limits organisms to grow. We added this information in the abstract to clarify our sentence.

3. Line 42. Are these MAGs coming only from the rhizosphere?

→ The MAGs collection was also constructed from bulk soil metagenomes. We constructed a non-redundant genome collection using dRep (Olm et al., 2017). This information was missing in the previous MS version. We have added these information in line 223 as follows “Maxbin2 MetaBAT2 and CONCOCT[57–59] were used to reconstruct metagenome-assembled genomes (MAGs) from bulk soil and rhizosphere samples. To dereplicate the MAGs, DASTool [60] was implemented. The quality of MAGs was calculated using CheckM [61]. Only medium quality MAGs according to the current definition of the minimum information metagenome-assembled genome (MIMAG) standards [62] and with at least 75% completeness were kept for further analyses. Metagenome-assembled genomes were then

dereplicated using dRep v2.2.3 (Olm et al., 2017) to obtain a non-redundant metagenome-assembled bacterial genome set.”

4. Line 99-103. While Lin et al 2022 describe the overall trend of the rhizosphere microbiome in terms of composition/assembly and predicted function/trophic behaviours, a work recently published confirmed the predicted functionality by applying the metagenome approach (see 10.1038/s43705-022-00130-7). The authors showed that the root system represents a microbial density and competition hotspot ruled out by a dual selection process: to colonize the plant-associated niches microorganisms must (1) possess the capacity to improve the fitness and survival of the plant (plant selection of beneficial microbes, PGP bacteria) and be able (2) to compete successfully against other microorganisms (microbial competition). It will be interesting to explore if this competition is also present in salty/desiccation soils and how this occurs along the desiccation gradient.

→ This is an interesting point. Evidence that microbes possess the capacity to improve the fitness and survival of the plant were shown based on genome centric analysis. We observed that many of the recovered MAGs harboured genes related to biochemical pathways that may improve plant growth (line 380) also protection against abiotic factors i.e., spermidine and betaine transport systems. We also extensively searched genes related to plant growth promotion and protection against abiotic factors in our metagenome dataset (see list of the genes below). The abundance profiles of these genes changed within the gradient of desiccation (line 338). For example, a high abundance of genes associated with osmoprotectant transporters increased along the desiccation gradient as well as genes involved in detoxification of heavy metals. Due the importance of referred paper regarding the recruitment of microbes with functional capacity to improve plant growth in desert environment, we added this reference into the manuscript (line 461).

“A recent study from a hyper arid environment also suggested that plant may have co-evolved with specific bacterial taxa that possess the capacity to improve the fitness and survival of the plant (Marasco et al., 2022).”

Table 1. List of genes that are associated with plant growth promotion and protection against biotic and abiotic stress detected in the metagenome dataset.

gene_id	gene_description
K07464	cas4; CRISPR-associated exonuclease Cas4 [EC:3.1.12.1]
K07725	csa3; CRISPR-associated protein Csa3
K19074	csa2; CRISPR-associated protein Csa2
K19075	cst2, cas7; CRISPR-associated protein Cst2
K19085	csa1; CRISPR-associated protein Csa1
K19086	csa4, cas8a2; CRISPR-associated protein Csa4
K19087	csa5; CRISPR-associated protein Csa5
K19088	cst1, cas8a; CRISPR-associated protein Cst1

K19089 cas5a_b_c; CRISPR-associated protein Cas5a/b/c
K19090 cas5t; CRISPR-associated protein Cas5t
K19091 cas6; CRISPR-associated endoribonuclease Cas6 [EC:3.1.-.-]
K07464 cas4; CRISPR-associated exonuclease Cas4 [EC:3.1.12.1]
K19114 csh1; CRISPR-associated protein Csh1
K19115 csh2; CRISPR-associated protein Csh2
K19116 cas5h; CRISPR-associated protein Cas5h
K07464 cas4; CRISPR-associated exonuclease Cas4 [EC:3.1.12.1]
K19117 csd1, cas8c; CRISPR-associated protein Csd1
K19118 csd2, cas7; CRISPR-associated protein Csd2
K19119 cas5d; CRISPR-associated protein Cas5d
K07464 cas4; CRISPR-associated exonuclease Cas4 [EC:3.1.12.1]
K19091 cas6; CRISPR-associated endoribonuclease Cas6 [EC:3.1.-.-]
K19120 csc1; CRISPR-associated protein Csc1
K19121 csc2; CRISPR-associated protein Csc2
K19122 csc3; CRISPR-associated protein Csc3
K19046 casB, cse2; CRISPR system Cascade subunit CasB
K19123 casA, cse1; CRISPR system Cascade subunit CasA
K19124 casC, cse4; CRISPR system Cascade subunit CasC
K19125 casD, cse5; CRISPR system Cascade subunit CasD
K19126 casE, cse3; CRISPR system Cascade subunit CasE
K19127 csy1; CRISPR-associated protein Csy1
K19128 csy2; CRISPR-associated protein Csy2
K19129 csy3; CRISPR-associated protein Csy3
K19130 csy4, cas6f; CRISPR-associated endonuclease Csy4 [EC:3.1.-.-]
K19131 csb1; CRISPR-associated protein Csb1
K19132 csb2; CRISPR-associated protein Csb2
K19133 csb3; CRISPR-associated protein Csb3
K19134 csx10; CRISPR-associated protein Csx10
K19135 csx14; CRISPR-associated protein Csx14
K19136 csx17; CRISPR-associated protein Csx17
K07012 cas3; CRISPR-associated endonuclease/helicase Cas3 [EC:3.1.-.- 3.6.4.-]
K07475 cas3; CRISPR-associated endonuclease Cas3-HD [EC:3.1.-.-]
K19137 csn2; CRISPR-associated protein Csn2
K07464 cas4; CRISPR-associated exonuclease Cas4 [EC:3.1.12.1]
K09952 csn1, cas9; CRISPR-associated endonuclease Csn1 [EC:3.1.-.-]
K09002 csm3; CRISPR-associated protein Csm3
K19138 csm2; CRISPR-associated protein Csm2
K19139 csm4; CRISPR-associated protein Csm4
K19140 csm5; CRISPR-associated protein Csm5
K07061 cmr1; CRISPR-associated protein Cmr1
K09000 cmr4; CRISPR-associated protein Cmr4
K09127 cmr3; CRISPR-associated protein Cmr3
K19076 cmr2, cas10; CRISPR-associated protein Cmr2
K19141 cmr5; CRISPR-associated protein Cmr5
K19142 cmr6; CRISPR-associated protein Cmr6
K19143 csx1; CRISPR-associated protein Csx1
K19144 csx3; CRISPR-associated protein Csx3
K19145 csx16; CRISPR-associated protein Csx16
K19146 csaX; CRISPR-associated protein CsaX
K07016 csm1, cas10; CRISPR-associated protein Csm1
K09951 cas2; CRISPR-associated protein Cas2
K15342 cas1; CRISPR-associated protein Cas1
K05813 ugpB; sn-glycerol 3-phosphate transport system substrate-binding protein
K05814 ugpA; sn-glycerol 3-phosphate transport system permease protein
K05815 ugpE; sn-glycerol 3-phosphate transport system permease protein
K05816 ugpC; sn-glycerol 3-phosphate transport system ATP-binding protein [EC:3.6.3.20]
EC:1.20.4.1 arsenate reductase (glutaredoxin); ArsC (ambiguous)
EC:1.20.99.1 arsenate reductase (donor); arsenate:(acceptor) oxidoreductase
EC:1.16.1.1 mercury(II) reductase
K17050 selenate/chlorate reductase subunit alpha[EC:1.97.1.9 1.97.1.1]
K00368 nir, nitrite reductase (NO-forming) [EC:1.7.2.1] [RN:R00783 R00785]

K00370 nxr, nitrite oxidoreductase [EC:1.7.99.4] [RN:R00798]
K00371 nxr, nitrite oxidoreductase [EC:1.7.99.4] [RN:R00798]
K10535 hox, hydroxylamine oxidase [EC:1.7.2.6] [RN:R10164]
K15864 nir, nitrite reductase (NO-forming) [EC:1.7.2.1] [RN:R00783 R00785]
K20932 hzs; Hydrazine synthase subunit (aka: hzsC) [EC:1.7.2.7] [RN:R09799]
K20933 hzs; Hydrazine synthase subunit (aka: hzsC) [EC:1.7.2.7] [RN:R09799]
K20934 hzs; Hydrazine synthase subunit (aka: hzsC) [EC:1.7.2.7] [RN:R09799]
K20935 hdh; hydrazine dehydrogenase
K20935 hzs; Hydrazine synthase subunit (aka: hzsC) [EC:1.7.2.7] [RN:R09799]
K00360 assimilatory nitrate reductase [EC:1.7.99.-] [RN:R00798]
K00366 assimilatory nitrite reductase [EC:1.7.7.1] [RN:R00790]
K00367 assimilatory nitrate reductase [EC:1.7.7.2] [RN:R00791]
K00372 assimilatory nitrate reductase [EC:1.7.99.-] [RN:R00798]
K10534 nitrate reductase (NAD(P)H) [EC:1.7.1.1 1.7.1.2 1.7.1.3] [RN:R00794 R00796]
K17877 assimilatory nitrite reductase [EC:1.7.1.4] [RN:R00787 R00789]
K00370 nitrite oxidoreductase [EC:1.7.99.4] [RN:R00798]
K00371 nitrite oxidoreductase [EC:1.7.99.4] [RN:R00798]
K10535 hydroxylamine oxidase [EC:1.7.2.6] [RN:R10164]
K10944 ammonia monooxygenase [EC:1.14.99.39] [RN:R00148]
K10945 ammonia monooxygenase [EC:1.14.99.39] [RN:R00148]
K10946 ammonia monooxygenase [EC:1.14.99.39] [RN:R00148]
K00368 nitrite reductase (NO-forming) [EC:1.7.2.1] [RN:R00783 R00785]
K00370 nitrate reductase 1 [EC:1.7.99.-] [RN:R00798]
K00371 nitrate reductase 1 [EC:1.7.99.-] [RN:R00798]
K00374 nitrate reductase 1 [EC:1.7.99.-] [RN:R00798]
K00376 nitrous-oxide reductase [EC:1.7.2.4] [RN:R02804]
K02305 nitric oxide reductase [EC:1.7.2.5] [RN:R00294]
K02567 periplasmic nitrate reductase NapA [EC:1.7.99.-] [RN:R00798]
K02568 cytochrome c-type protein NapB [RN:R00798]
K04561 nitric oxide reductase [EC:1.7.2.5] [RN:R00294]
K15864 nitrite reductase (NO-forming) [EC:1.7.2.1] [RN:R00783 R00785]
K15877 fungal nitric oxide reductase [EC:1.7.1.14] [RN:R02492 R09446 R09808 R09809]
K00362 respiratory nitrite reductase [EC:1.7.1.15] [RN:R00787]
K00363 respiratory nitrite reductase [EC:1.7.1.15] [RN:R00787]
K00370 respiratory nitrate reductase 1 [EC:1.7.99.-] [RN:R00798]
K00371 respiratory nitrate reductase 1 [EC:1.7.99.-] [RN:R00798]
K00374 respiratory nitrate reductase 1 [EC:1.7.99.-] [RN:R00798]
K02567 dissimilatory nitrate reductase [EC:1.7.99.-] [RN:R00798]
K02568 dissimilatory nitrate reductase [EC:1.7.99.-] [RN:R00798]
K03385 respiratory nitrite reductase [EC:1.7.2.2] [RN:R05712]
K15876 respiratory nitrite reductase [EC:1.7.2.2] [RN:R05712]
K02575 NRT; MFS transporter, NNP family, nitrate/nitrite transporter
K10535 hydroxylamine oxidase [EC:1.7.2.6] [RN:R10164]
K10944 ammonia monooxygenase [EC:1.14.99.39] [RN:R00148]
K10945 ammonia monooxygenase [EC:1.14.99.39] [RN:R00148]
K10946 ammonia monooxygenase [EC:1.14.99.39] [RN:R00148]
K20932 hzs; Hydrazine synthase subunit (aka: hzsC) [EC:1.7.2.7] [RN:R09799]
K20933 hzs; Hydrazine synthase subunit (aka: hzsC) [EC:1.7.2.7] [RN:R09799]
K20934 hzs; Hydrazine synthase subunit (aka: hzsC) [EC:1.7.2.7] [RN:R09799]
K20935 hzs; Hydrazine synthase subunit (aka: hzsC) [EC:1.7.2.7] [RN:R09799]
K00531 nitrogenase delta subunit [EC:1.18.6.1] [RN:R05185]
K02586 nitrogenase molybdenum-iron protein [EC:1.18.6.1] [RN:R05185]
K02588 nitrogenase iron protein [RN:R05185]
K02591 nitrogenase molybdenum-iron protein [EC:1.18.6.1] [RN:R05185]
K22896 vanadium-dependent nitrogenase [EC:1.18.6.2] [RN:R12084]
K22897 vanadium-dependent nitrogenase [EC:1.18.6.2] [RN:R12084]
K22898 vanadium nitrogenase delta subunit [EC:1.18.6.2] [RN:R12084]
K22899 vanadium nitrogenase iron protein [RN:R12084]
K00855 phosphoribulokinase [EC:2.7.1.19] [RN:R01523]
K01601 ribulose-bisphosphate carboxylase [EC:4.1.1.39] [RN:R00024]
K01602 ribulose-bisphosphate carboxylase [EC:4.1.1.39] [RN:R00024]
K00548 methionine synthase [EC:2.1.1.13 2.1.1.14] [RN:R00946 R04405]

K00549	methionine synthase [EC:2.1.1.13 2.1.1.14] [RN:R00946 R04405]
K05845	ABC.BCP1.S; osmoprotectant transport system substrate-binding protein
K05846	ABC.BCP1.P; osmoprotectant transport system permease protein
K05847	ABC.BCP1.A; osmoprotectant transport system ATP-binding protein
K02052	ABC.SP.A; putative spermidine/putrescine transport system ATP-binding protein
K02053	ABC.SP.P; putative spermidine/putrescine transport system permease protein
K02054	ABC.SP.P1; putative spermidine/putrescine transport system permease protein
K02055	ABC.SP.S; putative spermidine/putrescine transport system substrate-binding protein

We additionally searched for genes associated with antibiotic biosynthesis and calculated their total abundance (RPKM, Figure 1). However, we did not find significant differences between the sampling sites (Kruskal-Wallis - P-value = 0.429).

Figure 1. Abundance of genes that were associated to antibiotic biosynthesis.

5. Lines 108-109. Among others, 10.1038/s41396-022-01238-3 showed how the autochthonous community in desert soil promotes the growth of plants, suggesting the potential of edaphic microbes and their ability to further colonize new plants and exert their beneficial services. I suggest better discussing the functional aspect of soil community and interaction with plants.

→ We agree with the reviewer. This aspect was discussed in line 525 -535.

6. Line 130. How did you collect 5 g of rhizosphere? How many plants were used for each replicate? Is this species the main abundance? How is it distributed along the desiccation gradient?

→ From each area, samples were taken from three independent biological replicates (from three different *S. acuminata* populations), each consisting of combined roots of three individual plants to obtain at least 5 g material per biological replicate. Rhizosphere samples were collected by light shaking of the roots to remove loosely attached soil before they were further treated in the laboratory as described below.

To clarify our sampling, strategy, we revised our method as follows (line 121):

“From each area, samples were taken from three independent biological replicates (from three different *S. acuminata* populations), each consisting of combined roots of three individual plants to obtain at least 5 g material per biological replicate. Rhizosphere samples were collected by light shaking of the roots to remove loosely attached soil before they were further treated in the laboratory as described below.”

7. Line 131. How was plant species identified during the sampling? Did you perform a molecular check on all samples collected to confirm they are the same species? Are the plants selected in the same phenological stage and have the same size? Could the latest be affected by salinity/desiccation? This is also an interesting point to be explored/discussed/tested.

The plant species was identified by a plant botanist and all of the plants were the same size – we included this information now in the Methods. We did not perform any molecular identification of the plant species. Nevertheless, the plant morphology and description are matched to *S. acuminata*. We added a representative figure of *S. acuminata* in the Supplementary Figure S1A.

8. Lines 142-143. This passage is not clear. You added 20 ml of solution, but you stated that the resulting suspension was 2ml. Did you just collect a portion? Please clarify and add the range of soil used for DNA extraction (weighted after the centrifuge) so that readers have an idea.

→ We agree this passage is not clear. We only used a fraction of the total samples for the DNA extraction. We revised our sentence in line 136 as follows:

“Fractions of 2 mL of the resulting suspensions were centrifuged at 16,000 x g and 4 °C with a Sorvall RC-5B Refrigerated Superspeed Centrifuge (DuPont Instruments; USA) for 20 min.”

The weight of the pellet was approx. 0.1 g. We added this information to the text.

9. Lines 156-157. How were qPCR results normalized? Based on the taxonomy obtained the different copy numbers of 16S rRNA of bacterial and archaea communities could be retrieved across the three sites.

→ The qPCR data was normalized according to the weight of the samples. We added information on how the copy numbers were computed as below:

“Serial dilutions of a standard containing a defined 16S rRNA gene copy number of *Bacillus* sp. and *Haloferax denitrificans* were used for the calculation of bacterial and archaeal gene

copy numbers in different samples, respectively. The copy numbers of bacterial and archaeal 16S rRNA genes were normalized according to the weight of the samples.”

10. Line 179. Why Silva 128? A more recent version (138) is available. Is the classifier trained with the primer used? Please specify how many chloroplasts (and non-bacteria) were obtained and provide rarefaction curves. Same for metagenomes.

→ This was a mistake, we used Silva 132, however not Silva 138. We additionally performed a taxonomic assignment using the more recent version of Silva and observed that general statistical patterns i.e., beta diversity are congruent between the “older” and “most recent” Silva databases. Hence, we decided to keep it as it was.

11. Line 202. Is the dataset also normalized based on the number of the 16S rRNA gene present in different taxa/ASVs?

→ The data is only normalized on the kingdom level (*Bacteria* and *Archaea*). This approach has been used in other studies (Dreier et al., 2022; Tilston et al., 2018). By doing so, we coupled the quantification accuracy of the qPCR with the capacity of amplicon sequencing to describe complex microbial communities

12. Line 248 repetition of "dried out"

→ Thank you for pointing this out.

11. Line 240. The authors should briefly describe the condition of the three sites, and, if it is possible, also include C and other nutrient contents.

→ We have added a picture gallery of the sampling location and described it briefly in line 126-131 (Supplementary Figure 1).

13. Lines 268-269. I suggest keeping the definition of the gradient as "desiccation" and not temporal. What do you mean by "Temporal changes"? if all the plants and soil were collected on the same day, the main changes observed in the rhizosphere/soil across the three sites are due to the "desiccation gradient". Moreover, the same desiccation is correlated/explained by time, so only one of these factors can be used to describe/explain the differences observed.

→ We agree with the reviewer. We removed “temporal” throughout the MS.

15. Line 306. Remove highlight from "Glycomyces".

→ It is now removed.

16. Lines 327-333. Specify the type of genes included. Are all those differentially distributed across samples, or only those related to PGP/beneficial traits?

→ We mentioned that the majority of the genes that were differentially abundant across the samples were genes encoding enzymes with EC numbers and transporter proteins (line 340). We particularly focused on those that have roles in supporting plant growth i.e., osmoprotectant, denitrification, and converting toxic compounds, and observed that the abundance of these genes shifted along the desiccation gradient (line 338-365). We discussed it further in line 468-490.

17. Line 436. Why do the mechanisms describe here can be defined as new? Several works showed how the rhizosphere tends to select core functional microbiome, as well as microbes enriched in PGP functions/traits that support plant growth. If this statement refers only to the archaea/bacteria ratio, please clarify it.

→ We removed “novel” from the sentence.

18. Figure 1. specify from amplicon dataset as done in Figure 2.

→ Thank you for pointing this out. The figure legend is now revised.

19. Figure 4A. maybe it can be clustered based on the distribution of metabolism across MAGs or on the distribution of MAGs in the three sites. It is difficult to follow the pattern described.

→ We generated a new figure and included a bar plot to show MAG abundance distribution in the three sites.

Figure 4. Gene profiles for selected functions in MAGs recovered from the dried-out Aral Sea basin and clustering based on genes encoding transporter proteins from each MAG. Presence (green) /absence (blue) plots showing profiles of selected genes in MAGs recovered from the dried-out Aral Sea basin (A). Abundances of MAGs based on the number of mapped reads per kilobase per million reads (RPKM) ($n=3$ replicates) in the 5-years dry area (red), 10-years-dry area (green) or 40-years-dry area (blue) (B; detailed in Table S5). Principal coordinate analysis (PCoA) of MAG clusters is based on genes encoding transporter protein counts in the genomes (C). MAGs belonging to *Halomonas* are indicated with arrows (C).

20. Figure 5. Correlation with chemistry data already published could additionally explain the pattern observed, especially trophic behaviours and C-metabolisms. Structural equation modelling (SEM) could be useful to explain the pattern observed, as well as it could be interesting to explore the correlation between microbes involved in different carbon and nitrogen cycling at different sites (e.g., pathway and corresponding percentage of genomes encoding the respective genes by using METABOLIC doi:10.1186/s40168-021-01213-8). → This is an interesting point. However, such an analysis would need more data points. We are planning to perform a more comprehensive follow-up study and combine it with a culture-omics approach to reveal microbial functions involved in plant growth promotion and abiotic stress including carbon and nitrogen cycling.

21. Table S1. What does star (*) means? From the caption, it is not clear. I suppose that pools were performed for bulk soil samples. However, from the methods, it is not clear if this was done for both amplicon and metagenome sequencing.

→ The star means that the samples were pooled prior shotgun metagenome sequencing as mentioned in the methods (line 164-165). The passage in the methods was also revised as follows (line 166).

“Due to low DNA concentrations in the bulk soil samples, the triplicates had to be pooled for each soil location prior to library construction and shotgun metagenomic sequencing.”

21. Tables S2 and S3. Please specify that these tables refer to the amplicon datasets. What does star (*) means? From the caption, it is not clear.

→ Thank you for pointing these details out. The data was based on amplicon sequencing data. We revised the figure captions accordingly and added the explanation of the asterix.

22. Table S5. I suppose this comes from the rhizosphere. Please specify. I also suggest, since three replicates were analyzed, adding the standard deviation to the relative abundance reported. If it is not possible, specify that the three metagenomes were pooled/merged to obtain MAGs; if it is the case how is relative abundance calculated? How did you compare? Which statistical analysis was applied?

→ The MAGs collection was also constructed from bulk soil metagenomes. We also constructed a non-redundant genome collection using dRep (Olm et al., 2017). This information was missing in the previous MS version. We added the information in line 224 as follows:

“Maxbin2 MetaBAT2 and CONCOCT[57–59] were used to reconstruct metagenome-assembled genomes (MAGs) from bulk soil and rhizosphere samples. To dereplicate the MAGs, DASTool [60] was implemented. The quality of MAGs was calculated using CheckM [61]. Only medium quality MAGs according to the current definition of the minimum information metagenome-assembled genome (MIMAG) standards [62] and with at least 75% completeness were kept for further analyses. Metagenome-assembled genomes were then dereplicated using dRep v2.2.3 (Olm et al., 2017) to obtain a non-redundant metagenome-assembled bacterial genome set.”

The abundance was estimated by mapping the Illumina reads for each of the MAGs using CoverM to produce relative abundance estimates of each MAGs. We used a pairwise Wilcox test to statistically analyze the abundance profiles of each MAGs between different sampling sites. We mentioned this in line 236. Moreover, we added the standard deviation into Supplementary Table S5.

23. Supplementary Figure S1. It could be interesting to present another set of panels with the same results from metagenomes. As well as add rarefaction curves for amplicon and

metagenomes approaches, at least based on 16S rRNA diversity in the rhizosphere and soil across the three-time points.

→ We added a bar plot that shows the relative abundance of bacteria and archaea of the total prokaryotic reads as determined using shotgun metagenome sequencing and mentioned it in the main text (Supplementary Figure S2B) as follows (line 285)

“Comparable to the qPCR results, the ratio between the archaeal and the bacterial relative abundance as determined using amplicon sequencing gradually decreased in the rhizosphere along the gradient of desiccation (Supplementary Fig. S2A). A similar pattern was observed with the metagenome dataset (Supplementary Fig. S2B).“

Supplementary Figure S1. Relative abundance between bacteria and archaea as determined using amplicon sequencing (A) and shotgun metagenome sequencing.

We also added rarefaction curves for the amplicon analysis as suggested (Supplementary Fig. S2).

Rarefractions curves indicated that our sequencing depth was sufficient to capture overall bacterial and archaeal diversity.

Supplementary Figure S2. Rarefaction curve based on the number of bacterial (A) and archaeal (B) species as determined using amplicon sequencing.

24. Supplementary Figure S3. What was used to calculate Shannon diversity? Did you use the 16S rRNA genes, MAGs, or functional genes from the metagenome approach? Please specify. Moreover, why are shown four black dots per box plot if the rhizosphere samples are three per time-point? See NMDS and Table S1.

→ The Shannon index was calculated using observed number of species and their abundances that were generated using the shotgun metagenomic sequencing approach. We clarified this approach in line 264 as follows:

“The Shannon diversity index based on shotgun metagenomic sequencing approach also indicated a congruent result ($P=0.429$ and $P=0.732$, respectively; Supplementary Fig. S4).”

The fourth dot is mean standard deviation. We decided to remove the dot to avoid reader’s confusion.

25. Supplementary Figure S4. Are A and C referring to the compositional diversity of community based on the 16S rRNA gene from metagenome/MAGs? Which taxonomic level was considered? Which functions were included in B and D? Please specify. Supplementary Figure S5. From which compartment the MAGs showed were obtained? Rhizosphere and/or soil?

→ Figure S4A and B refer to the community structure based on 16S rRNA amplicon sequencing whereas C and D refer to gene profiles based on the shotgun metagenome sequencing. We revised the figure legends (which is now Supplementary Figure 5 due to the additional Supplementary Figure) to avoid confusions for reader as follows:

Supplementary Figure S5. NMDS plot showing clustering of bacterial and archaeal community structures (A and B) based on 16S rRNA amplicon sequencing and functioning – annotated gene profiles based on shotgun metagenome sequencing (C and D).

As previously mentioned The MAGs collection was constructed from bulk soil as well as rhizosphere metagenome which were dereplicated using dRep to create a non-redundant MAGs collection. We added this information to the figure legend.

“Supplementary Figure S6. Phylogenetic tree of non-redundant metagenome assembled genomes (MAGs). Different bacterial and archaeal taxa in the phylogenetic tree constructed MAGs are highlighted with different colours.”

October 29, 2022

Dr. Tomislav Cernava
Graz University of Technology
Institute of Environmental Biotechnology
Graz
Austria

Re: mSystems00739-22R1 (Function-based rhizosphere assembly along a gradient of desiccation in the former Aral Sea)

Dear Dr. Tomislav Cernava:

I am pleased to inform you that your manuscript has been accepted, and I am forwarding it to the ASM Journals Department for publication. For your reference, ASM Journals' address is given below. Before it can be scheduled for publication, your manuscript will be checked by the mSystems production staff to make sure that all elements meet the technical requirements for publication. They will contact you if anything needs to be revised before copyediting and production can begin. Otherwise, you will be notified when your proofs are ready to be viewed.

Publication Fees:

If you would like to submit a potential Featured Image, please email a file and a short legend to mSystems@asmusa.org. Please note that we can only consider images that (i) the authors created or own and (ii) have not been previously published. By submitting, you agree that the image can be used under the same terms as the published article. File requirements: square dimensions (4" x 4"), 300 dpi resolution, RGB colorspace, TIF file format.

We recognize that the video files can become quite large, and so to avoid quality loss ASM suggests sending the video file via <https://www.wetransfer.com/>. When you have a final version of the video and the still ready to share, please send it to mSystems staff at mSystems@asmusa.org.

Sincerely,

Karoline Faust
Editor, mSystems

Journals Department
Fig. S6: Accept
Table S2: Accept
Fig. S1: Accept
Fig. S4: Accept
Table S1: Accept
Fig. S2: Accept
Fig. S5: Accept
Table S4: Accept
Table S3: Accept
Fig. S3: Accept